# Spatial covariance analysis reveals the residue-by-residue thermodynamic contribution of variation to the CFTR fold

Frédéric Anglès[1], Chao Wang[1] & William E. Balch 🔟 [1✉]

Although the impact of genome variation on the thermodynamic properties of function on the protein fold has been studied in vitro, it remains a challenge to assign these relationships across the entire polypeptide sequence in vivo. Using the Gaussian process regression based principle of Spatial CoVariance, we globally assign on a residue-by-residue basis the biological thermodynamic properties that contribute to the functional fold of CFTR in the cell. We demonstrate the existence of a thermodynamically sensitive region of the CFTR fold involving the interface between NBD1 and ICL4 that contributes to its export from endoplasmic reticulum. At the cell surface a new set of residues contribute uniquely to the management of channel function. These results support a general 'quality assurance' view of global protein fold management as an SCV principle describing the differential pre- and post-ER residue interactions contributing to compartmentalization of the energetics of the protein fold for function. Our results set the stage for future analyses of the quality systems managing protein sequence-to-function-to-structure broadly encompassing genome design leading to protein function in complex cellular relationships responsible for diversity and fitness in biology in response to the environment.

[1] Scripps Research, Department of Molecular Medicine, 10550 North Torrey Pines Rd, La Jolla, CA 92037, USA. ✉email: webalch@scripps.edu

Achieving the native physiologic functional state of the protein fold in vivo[1] is a complex process that reflects the sum of the multidimensional information defined among other things by the genome sequence, translation speed, specific interactions, and the state of the proteostasis network that includes the cellular components modulating folding and degradation in response to the dynamic and variable environments of the cell and its surroundings. Genetic variation across the population differentially affects protein function in complex cell, tissue, and host environments, resulting in a range of functional to dysfunctional phenotypes that contribute to human disease[2–4]. The different phenotypes of each of these variants reflect the impact of maladaptive folding intermediates[5], variants that we have previously proposed can be used to understand the relationships between genome sequence information and the protein fold that shapes function[4,6,7].

By using variants as reporters of sequence-to-function-to-structure relationships[4], we have developed variation spatial profiling (VSP), a Gaussian process regression (GPR) based method that can analyze and predict how each residue defines the polypeptide chain sequence impacts the folding and function of any protein[4,6,7] (Fig. 1a). VSP is based on the principle of spatial covariance (SCV) that uses a sparse collection of known variants found in the world-wide population and their associated biological features to quantitatively assign functional value (protein traits) as a matrix to all uncharacterized residues spanning the polypeptide sequence with defined uncertainty (Fig. 1a) thereby defining the probable role of each residue in the WT-fold on a residue-by-residue basis.

Understanding the role of the protein management system in making decisions as to when to generate and maintain the fold for stability or degrade a protein in the cell, and how it links to protein function is of critical importance to our knowledge of the differential biology in health and disease impacting each cell type. In the eukaryotic cell exocytic pathway, chaperone-assisted protein folding and endoplasmic reticulum (ER) associated degradation (ERAD) have been shown to play an important role in triaging proteins for export by the COPII export machinery[8–11]. The ER has been

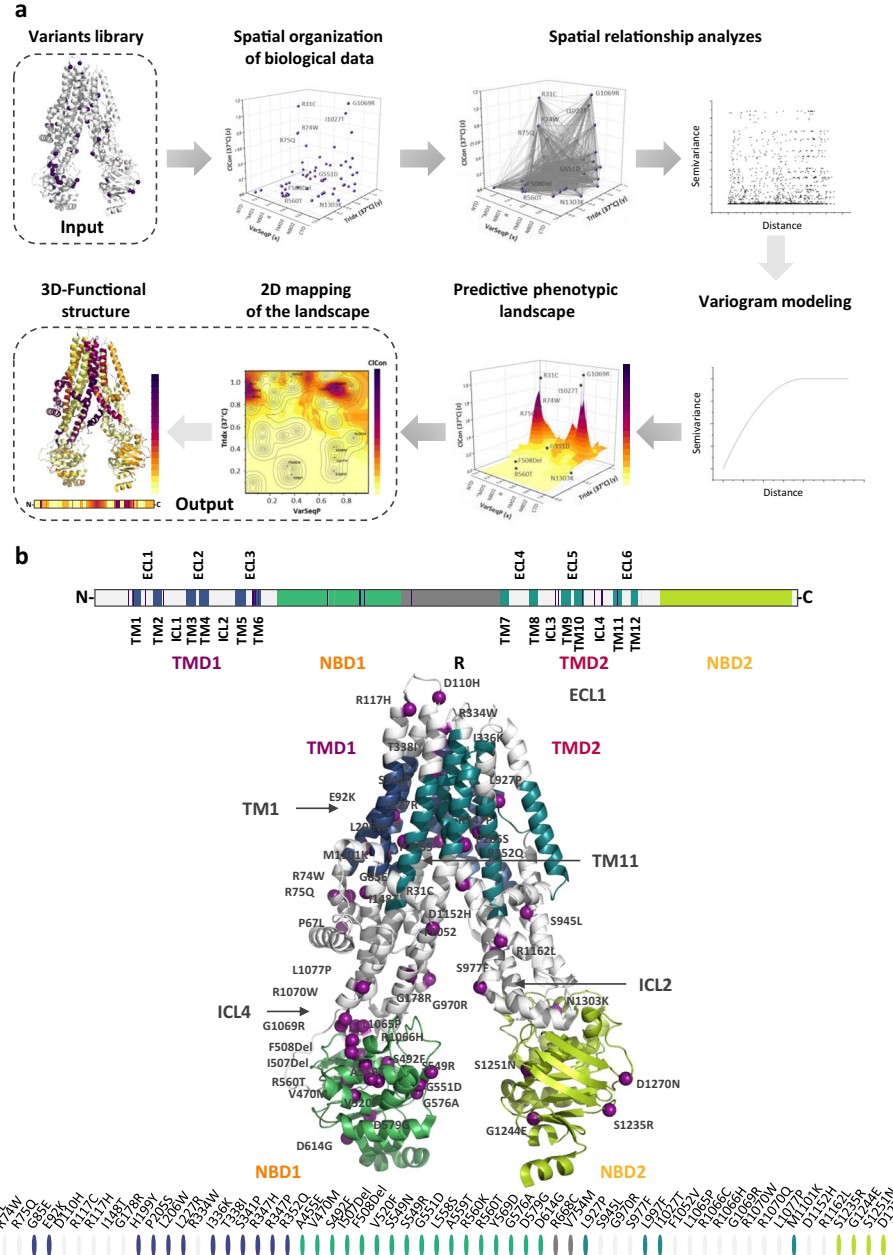

**Fig. 1 VSP workflow and CFTR organization. a** VSP workflow. Variant library: VSP uses a sparse collection of variants with known position and phenotypical traits. Spatial organization of the biological data: We position each of the variants on the *x*-axis by their normalized position in the linear polypeptide sequence, where the full-length chain is assigned a value of 1.0, referred to as the variant sequence position (VarSeqP). We assigned the value of each variant's feature (in this the TrIdx, see Results) to the *y*-axis. The trafficking index (TrIdx) is the normalized ratio of the fraction of variants exported from the ER relative to the total variant level in the cell, where the value of the WT CFTR TrIdx is assigned a value of 1.0. To correlate the sequence position and TrIdx values to CFTR chloride conductance (ClCon), the normalized ClCon is assigned to the *z*-coordinate, where the value of the WT CFTR ClCon is assigned a value of 1.0. Spatial relationship analysis: We calculate the 2D distance values based on VarSeqP and TrIdx, as well as the variance of ClCon for all possible 2080 (based on 64 variants) variant pairwise combinations. Variogram modeling: The molecular variogram shows the calculated spatial variance and distance values for each comparison. Such modeling quantitatively defines the linear sequence range where the variants co-vary with each other for a given set of functions defined by the *y*- and *z*-axis coordinates (TrIdx and ClCon, respectively) until it reaches a plateau that we refer to as the 'molecular range'. Variants with distance relationships extending beyond the range are generally not correlated[4]. Predictive phenotypic landscape: In the example shown, GPR uses the molecular variogram to predict all uncharacterized impacts of amino acid residue variation on the ClCon (*z*-axis) based on the residue position in the polypeptide sequence (*x*-axis) and the TrIdx (*y*-axis). 2D mapping of the 3D phenotype landscape: The 3D phenotype landscape is projected as a 2D heatmap of the *z*-coordinate ClCon values to generate a 2D phenotype landscape. A confidence interval (uncertainty) can be assigned to every data point that can be delineated by contour lines in the 2D projection map. 3D functional structure: We project the generated ClCon-phenotype landscape values with the highest confidence for all CFTR residues to create a ClCon functional structure displaying how amino acid variation at each position is impacting the functional-structure relationships. **b** Upper panel: domain organization of CFTR. Middle panel: visualization of the 64 variants (dark purple) used in the current study on the open (ATP-bound) structure of CFTR (PDB: 5UAK). Transmembrane domain (TMD)1 (purple), nucleotide-binding domain (NBD)1 (orange), R domain (RD), TMD2 (red) and NBD2 (yellow). Lower panel: the 64 listed variants used in the study are distributed as indicated by the domain color (ovals).

generically referred to as 'quality control (QC)' system[12–14] that only exports proteins with native conformation[13] competent for function in downstream compartments. However, this assumption is challenged in many diseases where the key proteins harbor multiple variants that have no effect on ER export yet can be completely dysfunctional in downstream compartments. An example is that of the cystic fibrosis transmembrane conductance regulator (CFTR) which, when mutated, contributes to CF. Whereas the F508del variant in the NBD1 domain (Fig. 1b) results in near total degradation by the ERAD[15–17], the G551D variant in the same domain as F508 is delivered with WT efficiently to the cell surface where it is functionally inactive as a chloride channel triggering severe disease[18]. These observations pose a major challenge in understanding how every residue contributes as part of a complex of direct and indirect physical-chemical interactions in the polypeptide chain that commit the fold to stability versus degradation, and ultimately to the native functional conformation(s).

To address the global features of managing the biological fold in vivo, we now apply VSP[4,6,7] to understand the thermodynamic design of the entire CFTR fold on a residue-by-residue basis that contributes to trafficking from the ER and chloride channel function at the cell surface. Using temperature-shift to study the energetics of the CFTR fold in vivo, we quantitatively assign the role of each residue of the CFTR polypeptide sequence required in different cellular compartments to achieve either stability, trafficking and/or function in the context of non-equilibrium biological thermodynamics[19,20]. We demonstrate the existence of a SCV defined thermodynamically sensitive region of the CFTR fold involving the interfaces between nucleotide-binding domain 1 (NBD1) and the fourth intracellular loop (ICL4) of transmembrane (TM) domain 2 (TMD2) is primarily responsible for the energetics of ER export[21–25]. At the cell surface a distinct set of residues contribute uniquely to the management of channel function. These results suggest an uncoupling of the energetics contributing to residue function for nascent synthesis, export, and trafficking versus downstream function at the cell surface- considerably expanding our understanding of triage in the living cell as dynamic collective system driving the physiological state of the fold.

We propose that the application of SCV principles[4] to understand the biological thermodynamic properties of the fold in vivo provides an universal quantitative tool to assess the conserved rules dictating covariant relationships fundamental to

functional protein design. We now propose an alternative view of triage in the cell from an energetic perspective as a more comprehensive 'quality assurance' (QA) program as a part of a larger quality system (QS)[26] that embraces the continuum of processes that ultimately contribute to the value of the functional protein fold in the cell.

## Results

**TrIdx and ClCon phenotypes of CFTR variants at 37 °C.** To address the role of energetics in managing the folding, stability, trafficking, and function of proteins traversing the exocytic pathway, we turned our attention to CFTR as a prototypical example of the impact of variation on protein fold and function trajectories given that ~2000 variants have been reported in the population with around 50% of them as missense mutations (www.genet.sickkids.on.ca; www.CFTR2.org)[3,27,28]. CF is a life-threatening Mendelian disease caused by genetic variation in the CFTR gene[3,4]. CFTR codes for an apically localized, chloride channel responsible for ion and fluid homeostasis of diverse tissues. The loss of a functional CFTR channel disrupts chloride ion homeostasis resulting in increased mucus viscosity in the airway of the lungs and ductal systems of the pancreas and liver, as well as decreased hydration of the intestinal tract[2,29]. Structurally, CFTR is composed of two nucleotide-binding domains (NBDs), a regulatory domain (R), and two transmembrane domains (TMDs), each composed of six transmembrane (TM) helices connected by extra- and intra-cellular loops (ECL and ICL, respectively)[30] (Fig. 1b). The biogenesis of CFTR involves nascent synthesis in the ER, the first step in the exocytic pathway, followed by trafficking through the Golgi to reach its destination at the apical cell surface of epithelial cells where it functions as a gated chloride channel[31,32].

To characterize the features of the fold contributing to the function of the CFTR polypeptide sequence, we measured at physiological temperature (37 °C) the trafficking and chloride conductance properties of 64 most frequent CFTR missense variants in the CF population that are dispersed along the entire polypeptide sequence dictating the organization of the structural fold[2–4] (Fig. 1b). Previous analyses of these variants revealed diverse functional relationships[2–4,33,34]. They include variants with a deficient trafficking and no chloride conductance (ClCon), as well as a wide range of trafficking values associated with partial to more normal ClCon. This library represents the most frequent

 3

variants in the CF population and is similar to the distribution of pathogenic/benign mutations of the >200 variants clinically characterized to date[35] (https://www.ncbi.nlm.nih.gov/clinvar/). Different from other mutation-based modeling approaches, VSP considers only mutations present in the population that are biologically relevant to onset and progression of human disease in order to focus on the key features of the fold that impact healthspan in the human population[4].

To calculate the trafficking index (TrIdx) of each variant, we transiently transfected HEK293 cells with each of the 64 variants in our library of native mutations responsible for the majority of CF in the clinic, and quantitated the relative abundance of the ER-localized core-glycosylated high mannose containing CFTR band B and the post-Golgi localized band C glycoforms that migrate uniquely on SDS-PAGE in response to trimming and modification by Golgi-localized enzymes (Supplementary Figs. 1–7, 8a upper panel)[36]. The TrIdx specifically represents the ratio of the post-Golgi localized band C glycoforms divided by the total CFTR amount (Band B + C). A comparison of the normalized TrIdx obtained from HEK293 cells (Supplementary Fig. 8a) to the TrIdx calculated from previously published data using Fischer rat thyroid (FRT) (Supplementary Fig. 8a) and HeLa cells[3,33] (Supplementary Fig. 8a) revealed a Pearson correlation coefficient (PCC) of 0.94 and 0.92, respectively (Supplementary Fig. 8b, c), suggesting that the TrIdx of common CFTR variants does not differ significantly between highly divergent heterologous cell lines reflecting conserved rules of protein design. The small difference between the datasets reported is likely due to a variable leakiness of variant export from the ER that is on average slightly higher in HEK293 cells when compared to the other cell lines.

To measure the ClCon of each variant at the cell surface, we adapted a fluorescence membrane potential (FMP) assay using a blue membrane potential dye previously described[37]. CFTR-mediated depolarization of the membrane is monitored as an increase in fluorescence. We normalized the fluorescence increase of each variant to that of WT-CFTR to obtain a normalized functional score for each CFTR variant (Supplementary Figs. 9–14). A comparison of the normalized function between HEK293 cells and FRT cells[3] (Supplementary Fig. 15a) yielded a PCC score of 0.80 (Supplementary Fig. 15b). In addition, while TrIdx studies of our variant library have not been previously reported in HBE cells, ClCon measurements for ~60 variants were previously performed in HBE cells[38]. The variants in common with our studies in HEK293 (R334W, I336K, T338I, S341P, F508Del, and G551D) behave in an identical fashion to what we have seen in HEK293 cells[38]. Furthermore, we transduced WT CFTR and 10 other variants (P67L, G85E, E92K, S492F, F508Del, G551D, R560T, L1077P, M1101K, and N1303K) in -/-CFBE41o- YFP cells (human bronchial epithelial cells not expressing CFTR) by adenovirus and observed a similar processing and activity of all tested variants in HEK293 and HBE cells (Supplementary Fig. 16). Together these results demonstrate that the basic functionality of diverse human variants spanning the entire sequence found in the population in different cellular environments is largely conserved.

**SCV relationships spanning the entire polypeptide chain**. To understand the relationships between TrIdx and ClCon across the entire polypeptide chain, we applied VSP, a GPR-based approach we have pioneered[4] that uses a sparse collection of variants distributed in the CF population to predict the full-spectrum of sequence-to-function-to-structure relationships based on the principle of SCV (Fig. 1a). The method relies on the GPR principle of weighted proximity, relating the sparse collection of variant positions in the sequence (a patient's genotype) to phenotypic traits, such as TrIdx and ClCon used

herein, to generate predicted values for the entire CFTR sequence[4]. To apply VSP, we position variants on the x-axis by their normalized position in the linear polypeptide sequence, where the full-length chain is assigned a value of '1', referred to as the variant sequence position (VarSeqP) (Fig. 1a). VarSeqP defines our first coordinate of a molecular spatial state encoded by the 1D sequence of the genome.

Given the importance of cellular location in CFTR along the exocytic pathway that contributes to function, we assigned the value of each variant's TrIdx (Supplementary Fig. 8a) to the y-axis (Fig. 1a). The TrIdx is the normalized ratio of the fraction of variants exported from the ER relative to the total variant level in the cell, where the value of the WT CFTR TrIdx is assigned a value of 1.0. The TrIdx provides a second coordinate in the molecular VSP strategy, thereby positioning the variant sequence information in a specified biological context.

To correlate the sequence position and TrIdx values to CFTR function at the cell surface (ClCon), the measured ClCon value for each variant is normalized to WT ClCon (set to 1.0) (Supplementary Fig. 15a). ClCon is assigned to the z-coordinate (Fig. 1a). To integrate sequence variation (VarSeqP) with biological features (TrIdx and ClCon) in 3D space, we calculate the 2D 'functional' distance values based on VarSeqP and TrIdx, as well as the variance of ClCon for all possible 2080 (based on 64 variants) variant pairwise combinations (Fig. 1a). These spatial relationships can be viewed and modeled in a molecular variogram (Fig. 1a)[4,39]. The molecular variogram defines the distance range based on the differences of variant sequence position and TrIdx value where the known variants co-vary with each other to generate the ClCon.

The molecular variogram provides a platform to apply GPR to predict amino acid function across the entire polypeptide sequence that can be projected as a 3D phenotype landscape (Fig. 1a). The 3D phenotype landscape reveals the covariant TrIdx and ClCon properties of the CFTR fold for each residue position in the context of the full-length polypeptide. When the 3D landscape (Fig. 1a) is projected as a 2D heatmap of the z-coordinate ClCon values (Figs. 1a and 2a), a confidence interval (uncertainty) that is assigned to every data point through GPR can be delineated by contour lines in the 2D projection map (Figs. 1a and 2a). The regions surrounded by more variants will have a higher confidence value (i.e., low uncertainty) associated with the prediction than the regions with fewer input variants in the landscape reflecting the natural distribution of variants in the population in the context of its evolving physiology defining disease in the population[4,6,39,40].

**SCV based functional structures**. To address sequence-to-function-to-structure SCV relationships in our CFTR example, we project the generated ClCon-phenotype landscape values with the highest confidence for all CFTR residues onto the structure of CFTR[30,41] to generate a ClCon 'functional structure' (Fig. 2a)[4]. The confidence value is strictly correlated with the variant distribution in the landscape, and the predictions with the highest confidence assigned to the functional-structure focus on the key features of the fold that are reported by genetic variants in the patient population defining evolutionary information contributing to CFTR fold function. Residues likely to convey values close to that of WT ClCon activity supporting normal lung function (>~0.5 of the ClCon value for WT found at 37 °C) are 'grayed' out, allowing us to focus on the effect of residues predicted to compromise function at 37 °C (Fig. 2b). The ClCon-phenotype landscape observed in HEK293 cells and the derived ClCon functional structure is similar to that observed in HeLa and FRT cells[4], an expected result given the high correlation for these

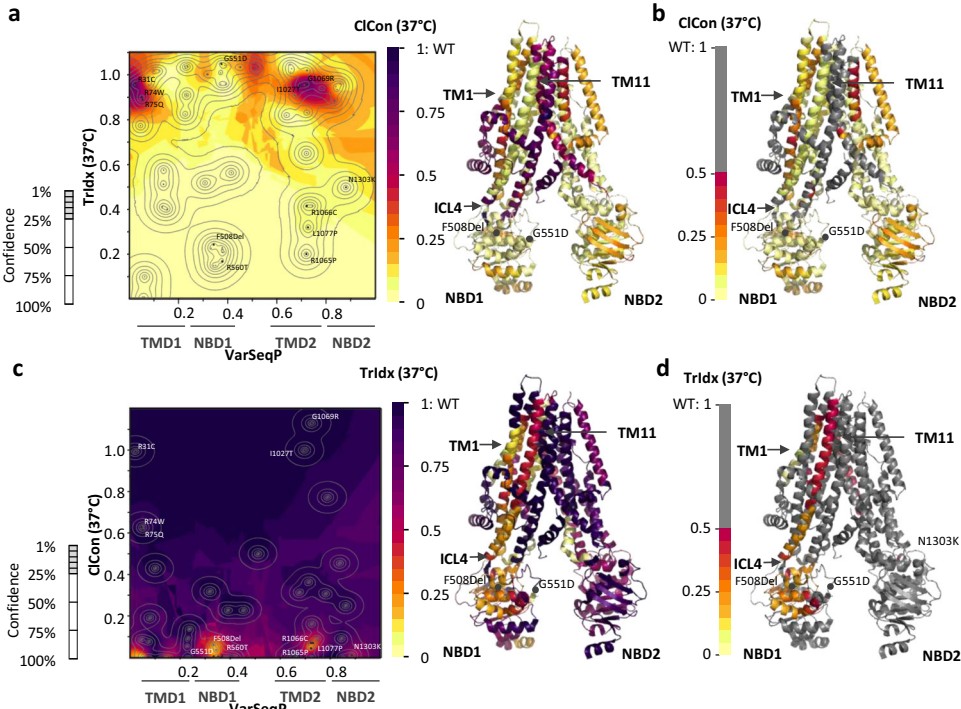

**Fig. 2 VSP-based ClCon- and TrIdx-phenotype landscapes at 37 °C. a** (Left panel) ClCon- phenotype landscape using the VSP-predicted values for TrIdx (band C/total (B + C) ratio; *y*-axis) and ClCon (*z*-axis) in response to sequence position (VarSeqP, *x*-axis). ClCon-phenotype landscape overlaid with 5, 10, 25, 50, and 75% uncertainty contour intervals (left panel, gray lines). Uncertainty values <25% are of high confidence and cover 95% of known and predicted residues. (Right panel) ClCon functional-structure (PDB: 5UAK) mapped using the highest confidence values for each residue found in the ClCon-phenotype landscape (left panel). **b** 'Grayout' of panel A ClCon functional-structure showing only residues with <50% of WT measured value. The grayout structure highlights in a color gradient the known and predicted unknown residue functions that are likely to contribute to disease due to reduced ClCon. **c** (Left panel) trafficking index (TrIdx)-phenotype landscape using the VSP-predicted values for ClCon (*y*-axis) and TrIdx (*z*-axis) in response to sequence position (VarSeqP, *x*-axis) overlaid with the confidence contour intervals. (Right panel) TrIdx-phenotype landscape defined functional structure (PDB: 5UAK). **d** 'Grayout' of panel C TrIdx functional structure showing only residues with <50% of WT measured value.

phenotype properties (Supplementary Fig. 15b). These results emphasize the conservation of CFTR fold when examining the fundamental SCV properties of trafficking from the ER and ClCon at the cell surface across divergent eukaryotic cell lines at 37 °C (Supplementary Figs. 8, 15).

While the ClCon-phenotype landscape conveys the contribution of each residue in the polypeptide chain to channel activity, by transposing the *y* and *z* axes found in the ClCon-phenotype landscape we can build a TrIdx-phenotype landscape (Fig. 2c) and the derived TrIdx functional-structure to display the contribution on the residue-by-residue basis the TrIdx of CFTR from the ER (Fig. 2c). We used a leave-one-out cross-validation (LOOCV) analysis given it is the best or among the best validation approaches for small sample size modeling[42] which yields a significant PCC of 0.57 and 0.79 with a $p = 5 \times 10^{-6}$ and $7 \times 10^{-12}$ (ANOVA test, with null hypothesis with the coefficient equal to zero), for ClCon-phenotype and the TrIdx-phenotype landscapes, respectively (Supplementary Data 1). To further validate the output of the ClCon- and TrIdx-phenotype landscapes (Fig. 2a, c), we used a different dataset of diverse CF variants[33,43] (Supplementary Fig. 17a) not included in the training dataset. These validation variants have diverse relationships between TrIdx and ClCon (Supplementary Fig. 17b, d). They include variants with deficient TrIdx and no ClCon, as well as a wide range of TrIdx values associated with partial to more normal ClCon (Supplementary Fig. 17a, b, d). The variants in the validation dataset are distributed across the entire CFTR sequence. Validation reveals a strong correlation (Supplementary Fig. 17c, e; PCCs of 0.67 (*p* value = $0.8 \times 10^{-3}$)

and 0.93 (*p* value = $1.1 \times 10^{-9}$) for ClCon and TrIdx, respectively) between all the experimentally measured values and the predicted values that define the output phenotype landscape.

We grayed out any value higher than 0.5 found in the TrIdx functional structure at 37 °C to focus on residues in the fold substantially contributing to ER export (Fig. 2d). The TrIdx functional structure shows that residues required for ER export are not distributed evenly throughout the polypeptide, but rather form specific clusters containing known and previously uncharacterized residues with severe trafficking defects. These residues form a region of the fold that contributes to the integrated TrIdx functional-structure properties of NBD1 with ICL4, TM11, and TM1 (Fig. 2c, d). The overall impact of residues on TrIdx is highly variable from modest to severe (Fig. 2c). For example, SCV reveals the critical role of TM1 on the TrIdx (Fig. 2c, d). TM1, the first membrane-spanning helix, has been shown to be important for the membrane integration of TMD1[44]. Furthermore, both ICL4 and TM11 in TMD2 have been shown to be essential for the assembly of the fully folded and functionally exportable CFTR protein[4].

Known and predicted residues with mild to no trafficking defects in the NBD2 domain reflect the fact that the folding of NBD2 is first influenced by the assembly of the NBD1-TMD1 (TM1) and TMD2 (ICL4/TM11) ER export sensitive residues[45,46] (Fig. 2c, d). This observation is consistent with the fact CFTR with NBD2 removed matures and has similar stability as the full-length WT CFTR leading us to conclude that catalytically crucial inter-NBD associations from a global SCV perspective are not required for ER export[47]. It is interesting that among the 64 variants used in the

study, the TrIdx of N1303K is the least similar amongst all cell lines tested to date (0.5 of WT in HEK293 cells compared to 0.1 of WT in HeLa cell[48] or HBE cells (Supplementary Fig. 16)). These results suggest that the folding and functional properties of NBD2 encompassing the N1303K SCV cluster could be differentially modulated by the specific biological properties of the tissue in which it is expressed[3,33,34,49–51].

Thus, in general, SCV relationships dictating the basic functional properties of CFTR appear similar across all cell lines, with few exceptions that likely reflect activities tailored to specific tissue environments, suggesting that unifying thermodynamic-based SCV principles are driving universal features of the functional fold in vivo.

**Biological thermodynamic SCV relationships dictating ER export.** Approximately 90% of the patient population carries at least one copy of the phenylalanine deletion at position 508 (F508del) in both homozygous and heterozygous states[2,3,34]. The F508del variant disrupts the proper folding of CFTR leading to its failure to be exported from the ER and clearance by ER-associated degradation (ERAD)[52–54]. While the impact of F508del on the thermodynamics of the NBD1 domain fold has been studied in vitro[55–58] and the energetic response of the F508del variant has been captured by temperature shift (37–27 °C) in vivo[16,59,60], it remains completely unknown how reduced temperature impacts the entire CFTR polypeptide sequence on a residue-by-residue basis as a SCV based matrix, a more global paradigm for understanding the thermodynamics of protein folding in vivo.

To address the role of energetics in CFTR fold synthesis, stability, and function-to-structure relationships using SCV, we transiently expressed WT-CFTR and each of the 64 variants and monitored the steady-state distribution of the band B and C glycoforms using SDS-PAGE following incubation at 27 °C for 24 h (Fig. 3a and Supplementary Figs. 1–7). To compare the responsiveness of the variants to reduced temperature, we calculated the WT-normalized TrIdx for each variant at 37 °C (orange) and 27 °C (purple) (Fig. 3a). Surprisingly, most variants tested appear to be refractory to reduced temperature correction (Fig. 3a). Those variants with a WT-normalized TrIdx below 0.4 exhibited improved yet highly variable trafficking responses to reduced temperature. For example, in contrast to F508del CFTR, which is the most highly responsive non-trafficking variant at 37 °C to reduced temperature (Fig. 3a), the NBD1 S492F-CFTR variant that exhibits a defect of TrIdx of equal severity to that of the F508del variant at 37 °C, showed little TrIdx improvement at 27 °C (Fig. 3a). This observation was validated in HBE cells where S492F does not respond to temperature shift as well (Supplementary Fig. 16a). These results suggest that S492F may confer a dominant-negative effect on the thermodynamics of the fold, for example, potentially energetically destabilizing the folded state of NBD1 thereby preventing ER export[61]. Residues outside of the NBD1 domain that showed trafficking defects largely failed to be temperature corrected for export from the ER (Fig. 3a).

To expand the analysis of the effect of reduced temperature on select variants to the entire fold, we applied VSP to generate a temperature sensitive (TS) delta (Δ) TrIdx-phenotype landscape. To build the landscape we used as input variant positions on the primary amino acid sequence (VarSeqP; $x$-axis), the TrIdx at 37 °C ($y$-axis) and the TrIdx delta response between 27 °C and 37 °C ($z$-axis) of the 64 variants (Fig. 3b). The TSΔTrIdx-phenotype landscape (Fig. 3b) shows the temperature correct-ability (the delta response of TrIdx between 27 °C and 37 °C; $z$-axis) on a residue-by-residue basis ($x$-axis) in response to the TrIdx at 37 °C ($y$-axis). Using the highest confidence response for each residue, the TSΔTrIdx-phenotype landscape was mapped to

the CFTR structure to generate a TSΔTrIdx functional structure (Fig. 3b). We observed that variants having a TrIdx > 0.4 (Fig. 3b) exhibited less than a 20% increase in their TrIdx for ER export upon incubation at 27 °C suggesting they are already thermo-dynamically favored for efficient export despite having disease-causing function at the cell surface. Indeed, most residues were not responsive to reduced temperature with the exception of three regions—two in the NBD1 (region I & II) and one in the TMD2 domain comprising ICL4 (region III) (Fig. 3b), possibly reflecting energetic coupling between these domains.

To focus on energetic coupling, we generated a gray-out view of the TSΔTrIdx functional structure (Fig. 3c) which shows in color only those residues having a >50% response to temperature shift (Fig. 3c). The gray-out models reveal that clusters defining regions I-III converge in the context of the entire CFTR fold design at the NBD1-ICL4 interface (Fig. 3c). For example, variants with the highest TrIdx correction in response to reduced temperature, including F508 and R560 (NBD1), L1065 (ICL4) (Fig. 3c), show a critical role for these residues in stabilizing the thermodynamics of the fold to promote ER export and/or to prevent CFTR targeting for degradation. In the ICL4, only the residues 1061–1065, found directly at the interface with NBD1, are responsive to temperature shift. All other residues are less prone to variability in response to temperature shift (Fig. 3c and Supplementary Data 2). These results are consistent with the observation that residues 563–567 found at the ICL4 interface function as a di-acidic motif to confer COPII-dependent export from the ER as previously reported[21,22,62,63] (Fig. 3d). These conclusions are consistent with mass spectrometry analysis of residues undergoing methylation and ubiquitination where ubiquitination of di-acidic motif residues targets CFTR for degradation when challenged through misfolding by F508del[64].

Thus, thermodynamic sensitive SCV relationships that capture the energetic properties across the entire CFTR nascent fold in response to temperature shift in vivo reveals an integrated thermodynamic sensitive region comprising the NBD1-ICL4 interface that dictates SCV tolerance[4] of the fold for either recognition by the COPII vesicle machinery for export[11,23–25] or for degradation through ERAD[52,65–69] at the first step of the exocytic pathway.

**SCV relationships defining cell surface stability.** To assess the biological function-based thermodynamic properties of the CFTR fold at the PM using VSP, each variant was expressed at 37 °C and subsequently incubated at 27 °C for 24 h to promote stabilization and export of all reduced temperature-responsive variants to the PM (Fig. 4a). The cells were then returned to physiological temperature (37 °C) in the presence of cycloheximide (CHX) to block de novo protein synthesis[70] (Fig. 4a). To measure the post-ER stability of expressed CFTR variants, we calculated the % of band C remaining after a 120 min chase in the presence of CHX at 37 °C and normalized this percentage to that observed for WT-CFTR to obtain a normalized band C stability score (Fig. 4a and Supplementary Figs. 1–7). The variants were ordered from the most stable to the least stable at 37 °C with the L206W and R560T variants serving as examples of variants with WT-like stability and those that show significant destabilization, respectively. We observed that ~50% of the tested variants exhibited stability of band C that was less than that of WT-CFTR (Fig. 4a).

Using the TrIdx as the $y$-axis and the band C stability score as the $z$-axis values in our VSP analysis, we were able to build a stability band C (StabC)-phenotype landscape (Fig. 4b) that can predict the role of all residues with uncertainty impacting the post-ER thermodynamic stability of the CFTR polypeptide. These predictions were used to generate a StabC functional structure (Fig. 4b, c). Post-ER stability of CFTR variants is highly correlated

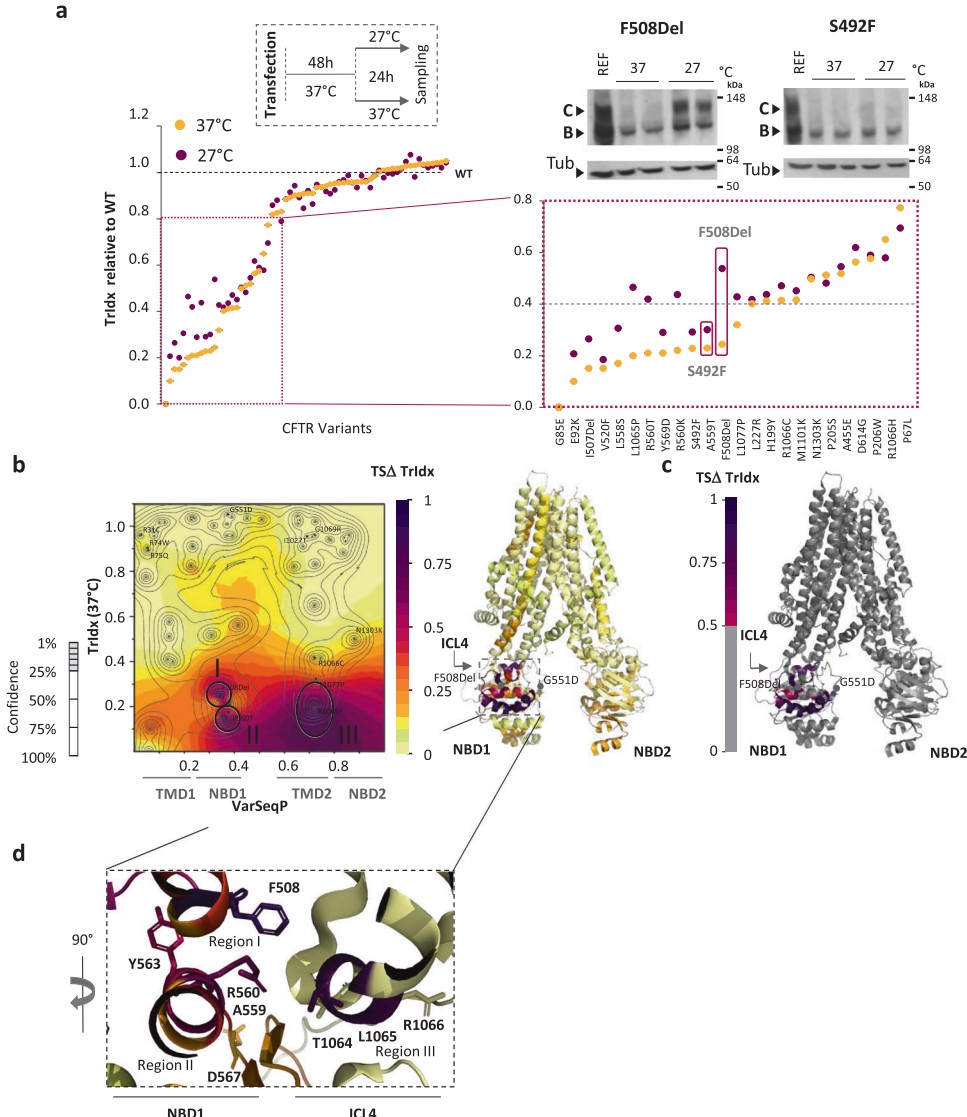

**Fig. 3 SCV relationships contributing to the energetics of ER export and trafficking. a** TrIdx (band C/total (B + C) ratio) of variants relative to WT at 37 °C (orange) and 27 °C (purple) (upper left inset). Variants are ordered from the lowest to the highest TrIdx at 37 °C (left panel). Zoom view of variants with trafficking defects <0.8 of WT value (right lower panel). SDS-PAGE of F508del and S492F variants after 24 h at 37 °C or 27 °C showing the difference between F508del (rescued) and S492F (not rescued) in response to reduced temperature (upper right insets). The two lanes per condition represent biological replicates. **b** (Left panel) SCV predicted temperature shift (TS) delta (ΔTrIdx (TSΔTrIdx) phenotype landscape using the VSP-predicted TrIdx at 37 °C (y-axis) values of the ΔTrIdx between 27 °C and 37 °C (z-axis) in response to sequence position (VarSeqP, x-axis) overlaid with the confidence contour intervals (left panel). (Right panel) The TSΔTrIdx-phenotype landscape defined functional structure. **c** Grayout of panel B TSΔTrIdx functional structure showing only residues with more than 50% of relative correction. The grayout structure highlights in a color gradient the known and predicted unknown residue functions that are likely to display TrIdx correction in response to reduced temperature. **d** Close-up view of the TrIdx correctability at the interface formed between NBD1/ICL4 including F508, Y563, R560, and D567 (NBD1) and T1064, R1066 (ICL4) in the TSΔTrIdx functional structure.

with their TrIdx at 37 °C (Supplementary Fig. 20). Variants with a TrIdx >0.5 of WT-CFTR all had a band C stability score of at least 0.5 of that seen with WT-CFTR (Fig. 4b). Furthermore, we noted that select variants at any known or predicted residue based on SCV in the first half of the protein (NT, TMD1, NBD1, R domains) that reduced the TrIdx of CFTR at 37 °C (TrIdx < 0.5), also reduced the post-ER stability of the polypeptide. These results illustrate that the SCV core linking NBD1 to TM1 and TM11 that determines the efficiency of ER export (Fig. 2d) can contribute to cell surface stability (Fig. 4c). These results suggest that energetic features used to maintain the function of the protein fold can be independent of subcellular location.

In general, the impact of NBD2 variants on protein stability is more dependent than NBD1 on localization, possibly reflecting

features of the genome to proteome design SCV relationships that contribute to differential function in distinct subcellular environments. These results stress the importance of assignment on a residue-by-residue basis through SCV in understanding management of the fold. Below we assess the impact of energetics on ClCon to correlate divergent requirements of sequence-to-function-to-structure design that link trafficking to channel function.

**Thermodynamic SCV relationships driving ClCon**. To assess the impact of energetics on ClCon, cells expressing the different variants were incubated at 27 °C for 24 h, then ClCon was measured immediately following shift to 37°. The activity of each

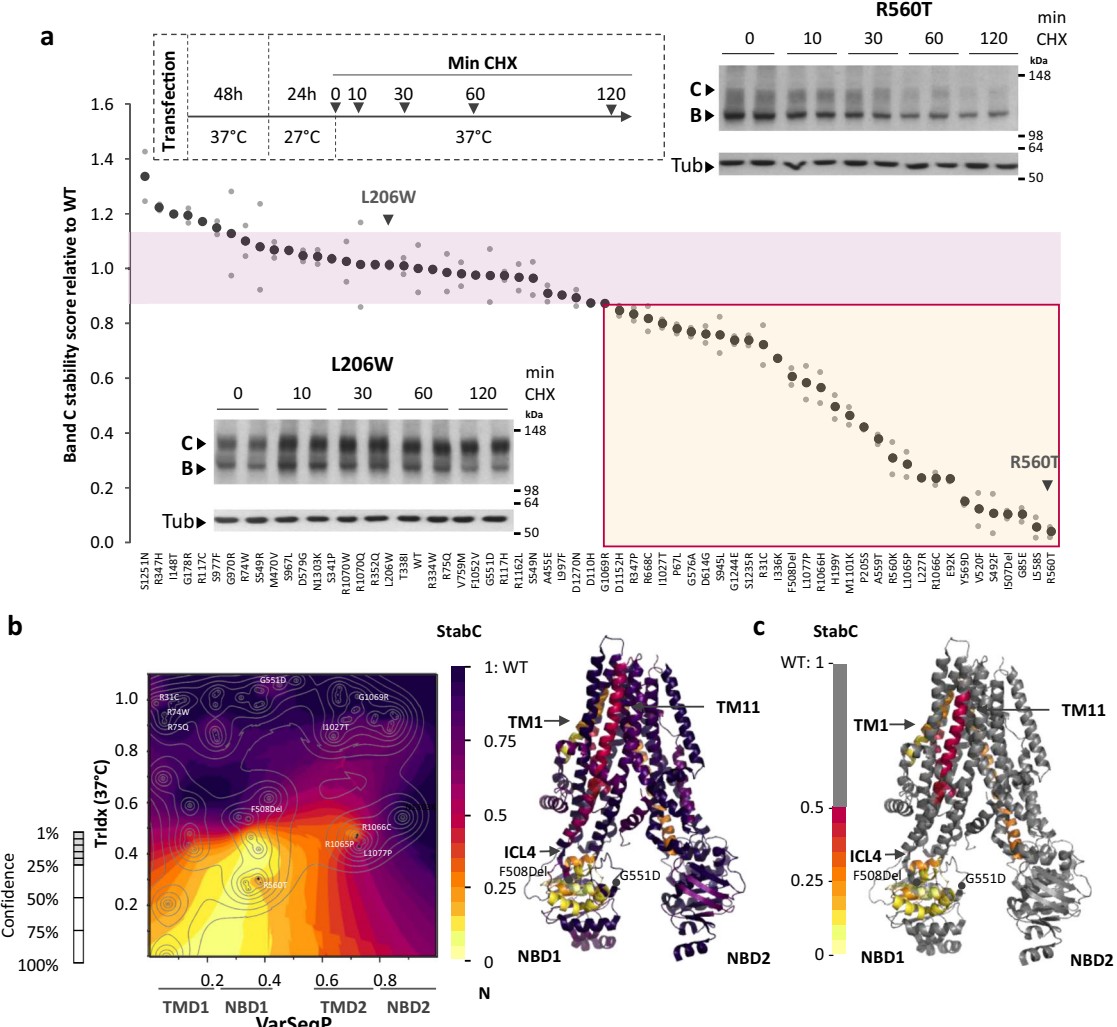

**Fig. 4 SCV relationships defining stability at the cell surface. a** The % of band C stability relative to WT of each variant at 37 °C was assessed after 24 h incubation at 27 °C followed by 120 min at 37 °C in the presence of CHX to inhibit protein synthesis ($n = 2$ biological replicates). Variants are ordered from the most to the least stable. The red box highlights variants that are destabilized relative to WT stability. SDS-PAGE of L206W (lower left insert) and R560T variants (upper right insert) following treatment with CHX for 0,10, 30, 60, and 120 min are shown as examples of WT-like stability and low-stability variants, respectively. **b** (Left panel) Stability of band C (StabC)-phenotype landscape using the VSP-predicted values of band C stability relative to WT ($z$-axis) relating to TrIdx (band C/total (B + C) ratio; $y$ axis) to the sequence position (VarSeqP, $x$-axis) overlaid with the confidence contour intervals. (Right panel) The StabC-phenotype landscape defined functional structure. **c** Grayout of panel B, StabC functional structure showing only residues with <50% of WT measured value. The grayout structure highlights in a color gradient the known and predicted unknown residue functions that are likely to contribute to disease due to reduced stability at the cell surface.

variant was normalized to that of WT-CFTR at 37 °C (Fig. 5a). We observed a striking variant-specific diversity of ClCon responses to reduced temperature pre-incubation. Some residues, including S1251N in NBD2 (Fig. 5a) failed to respond, while others including F508del, exhibited a significant improvement in channel activity at 27 °C (Fig. 5a and Supplementary Figs. 9–14).

To understand the relationships between TrIdx and ClCon at 27 °C for all residues contributing to the CFTR fold, we built a ClCon-phenotype landscape at 27 °C based on the TrIdx at 27 °C (Fig. 5b). The molecular variogram analysis (see Methods) when compared with that observed for ClCon-phenotype landscape at 37 °C based on the TrIdx at 37 °C (Fig. 2a and Supplementary Figs. 1–7) reveals that the functional spatial distance reflecting the difference in TrIdx measurement is significantly reduced by low temperature (Supplementary Figs. 1–7), consistent with the fact that a subset of variants with deficient trafficking were corrected by low temperature (Fig. 3a). In contrast, the diversity in

measured ClCon values is not reduced by low temperature with the mean of spatial variance of ClCon for all the pairwise comparisons of input variants under 27 °C even slightly higher than 37 °C (Supplementary Figs. 1–7). These results are in alignment with the highly variable improvement of ClCon observed in CF variants in response to low temperature (Fig. 5a). These results suggest that distinct thermodynamic sensitive mechanism(s) impact TrIdx and ClCon in response to temperature shift. The ClCon-phenotype landscape and the corresponding functional structure observed at 27 °C (Fig. 5b, c) generally have higher predicted ClCon values when compared with that at 37 °C (Fig. 2a, b), suggesting that the reduced temperature restoring ER export of temperature-sensitive residues can also restore channel activity.

To gain further insight into the coupling between TrIdx and ClCon, we examined the relationship of TrIdx at 37 °C to the Δ of ClCon achieved upon temperature shift. For this purpose, we

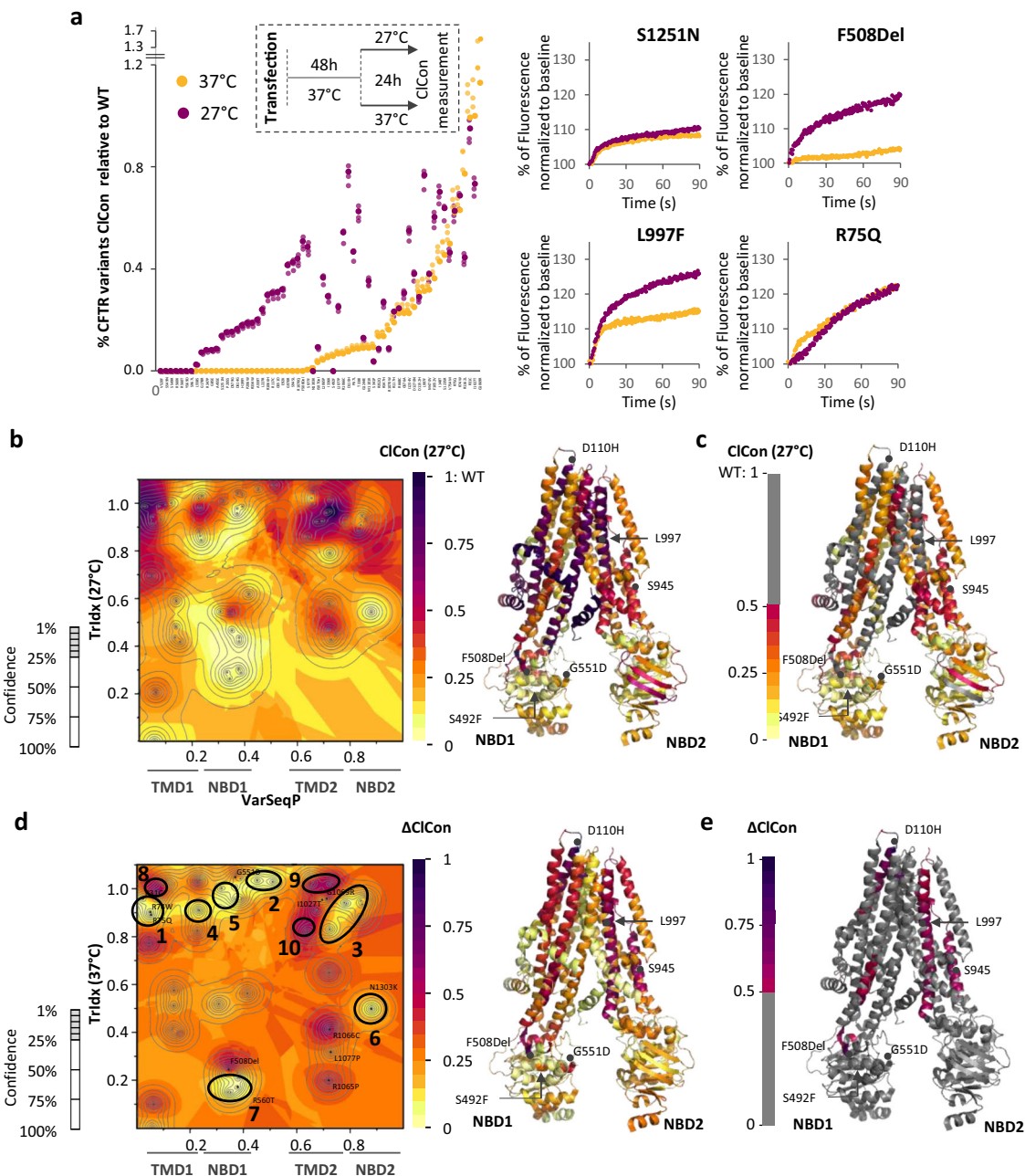

**Fig. 5 Contribution of SCV to CFTR energetics driving function at the cell surface. a** % of variant function relative to WT at 37 °C (orange) and 27 °C (blue); the variants are ordered from the lowest to the highest function measured at 37 °C (left panel; n = 3 biological replicates). Representative FMP traces of S1251N, F508Del, L997F and R75Q variants after 24 h at 37 °C (orange) or 27 °C (blue) (right panel). **b** (Left panel) ClCon-phenotype landscape at 27 °C using the VSP-predicted values relating TrIdx (band C/total (B + C) ratio; y-axis) and ClCon (z-axis) in the context of sequence position (VarSeqP, x-axis) overlaid with the confidence contour intervals. (Right panel) ClCon-associated functional structure at 27 °C. **c** Grayout of panel B, ClCon functional structure at 27 °C showing only residues with <50% of WT measured value. The grayout structure highlights in a color gradient the known and predicted unknown residue functions that are likely to contribute to reduced ClCon at 27 °C. **d** (Left panel) Δ of the temperature shift of ClCon (TSΔClCon) phenotype landscape using the VSP-predicted values of the Δ ClCon between 27 °C and 37 °C (z axis) relating to the TrIdx at 37 °C in the context of sequence position (VarSeqP, x-axis) overlaid with the confidence contour intervals. (Right panel) TSΔClCon-phenotype landscape defined functional structure. **e** Grayout of panel B, TSΔClCon functional structure showing only residues with more than 50% of relative correction. Maximal ΔClCon (70% of WT) is set as 1.0 (Fig. 5d, e). The grayout structure highlights in a color gradient the known and predicted unknown residue functions that are likely to display ClCon correction in response to reduced temperature.

used the TrIdx at 37 °C (Fig. 5d) and the Δ temperature shift of ClCon (TSΔClCon) reflecting rescue of function upon shift from 37 °C to 27 °C (Fig. 5d) to generate a TSΔClCon-phenotype landscape. The computed values were converted to the predicted

TSΔClCon functional structure (Fig. 5d) and its gray-out model (Fig. 5e) that highlights only those residues with >50% maximal Δ ClCon in response to temperature shift. Here, we see that many residues in the functional structure exhibit a >50% of maximal Δ

ClCon in response to reduced temperature (Fig. 5e, red to purple), a response that is largely independent of their trafficking ability (Fig. 5d). Furthermore, the TSΔClCon functional structure (Fig. 5e) is dramatically different from the TSΔTrIdx functional structure (Fig. 3c). These results suggest that reduced temperature impacts the SCV relationships of the CFTR polypeptide fold beyond simply stabilizing the fold for exit from the ER[5,16,17,21,60].

In contrast to SCV predicted regions of the fold that show improved ClCon in response to reduced temperature, a number of regions have little to no change reflecting a minimal impact on biologically relevant thermodynamic features of the fold (Fig. 5d). Clusters 1–3 encompass regions of the CFTR polypeptide where residues have near WT trafficking and a minimal effect on channel activity at 37 °C. Exposure to reduced temperature conditions does not improve their near WT activity, yielding a low TSΔClCon score (Fig. 5d). Moreover, clusters 4–7 encompass regions of the CFTR polypeptide fold where residues have no activity at physiological temperature (Supplementary Figs. 9–15) and do not respond to temperature shift to 27 °C (Fig. 5d). The combined results revealed by clusters 1–7 likely reflect the differential limits of thermodynamic plasticity of the fold.

Whereas non-responsive regions characterize a majority of the fold, we see specific clusters where residues exhibit improved channel activity upon temperature shift. For example, a region defined by cluster 8, which is located proximally to D110H variant at the N-terminus (N) that traffics normally at 37 °C (Fig. 1a, b), exhibits a weak basal ClCon that is dramatically increased in response to reduced temperature. Moreover, clusters 9 and 10 which are localized around measured L997 and S945 positions in the TMD2 domain, respectively, exhibit little to no activity at 37 °C (Fig. 5a; Supplementary Figs. 9–14) yet are responsive to reduced temperature correction of ClCon (Fig. 5d).

In general, SCV relationships capturing features of the fold responsive to temperature shift demonstrate that reduced temperature exposure differentially affects the biological thermodynamic properties of the CFTR polypeptide on a residue-by-residue basis that extends well beyond stabilization of the NBD1-ICL4 interface required to export CFTR from the ER. These results emphasize the differential plasticity of thermodynamic relationships dictating genome-based fold design of CFTR for ER export versus differential downstream function in post-ER compartments in response to its local folding and interaction environment[64,71] indicating the limitations of the QC concept.

**Functional thermostability in vivo is poorly correlated with in silico thermostability predictions.** Given the results of the temperature shift-based phenotype landscapes projecting the folding energetics linked to CFTR TrIdx and ClCon, we examined the relationship between these fundamental energetic features dictating fold function in vivo to energetic thermostability values (ΔΔG°) calculated for each variant included in the study using the thermostability predictive program FoldX[72–74]. FoldX calculates the ΔΔG° of a residue based on force field algorithm in response to a specific variation to capture the overall thermostability of protein structure (Supplementary Fig. 18a). Strikingly, comparing the FoldX predicted ΔΔG° between WT-CFTR and each of the CFTR variants included in the study with either the measured TrIdx (Supplementary Fig. 8a) or ClCon (Supplementary Fig. 15a) at 37° in vivo revealed that neither stability of the fold for exit from the ER as measured by the TrIdx at 37 °C (Supplementary Fig. 18b) nor function at the cell surface as measured by ClCon at 37 °C (Supplementary Fig. 18c) correlate with in silico predicted folding stability using FoldX[72–74] likely reflecting the non-equilibrium thermodynamic processes defining protein folding in physiologically relevant cellular environments.

**SCV based energetic code for the CFTR fold.** To address the integrated functional non-equilibrium thermodynamic differences driving CFTR function in health and disease on a residue-by-residue basis in response to a temperature shift, we first normalized the predicted ΔTrIdx (Fig. 3b) and ΔClCon (Fig. 5d) values for each residue relative to WT (Fig. 6a), and then calculated the difference between the normalized ΔTrIdx and ΔClCon values to generate a 'differential energetics based correction' (DEC) plot (Fig. 6a and Supplementary Data 2). The DEC plot allows us to codify on a residue-by-residue basis regions in the fold that are differentially contributing to the non-equilibrium energetics that are critical for the proper trafficking of CFTR from the ER and/or ability to achieve channel function at the cell surface. For example, most regions display a negative DEC value (Fig. 6a) reflecting the fact that they have a high ΔClCon value, yet a low ΔTrIdx in response to temperature shift (Fig. 6a). The DEC plot allows us to determine if the correction observed in the ER is causally correlated with the correction of ClCon at the cell surface. For example, the increase of 26% of WT level of TrIdx in response to the temperature shift observed in the ICL4 region between the position 1061 and 1065 is associated with a similar ClCon correction which results in a DEC value close to 0 (Fig. 6a and Supplementary Data 2). We observe a similar effect for the region contributing to the function of the F508 residue (Fig. 6a and Supplementary Data 2), consistent with the observation that single-channel patch-clamp experiments reveal that temperature shift has no effect on current flow through the F508del channel in the full open state[75] but can modulate the frequency and duration of F508del-CFTR channel opening with a maximum open probability at 30 °C[76]. While we did not monitor single-channel activity nor quantify the absolute number of channels at the cell surface, SCV analyses suggest that the increased ClCon in response to F508 and the region define by 1061 to 1065 is largely due to the amount of variant trafficked to the surface (Fig. 6a, b and Supplementary Data 2). In contrast, SCV relationships dictated by residues between 512–536 and the di-acidic exit motif (YKDAD (residues 563–567)) have a strongly positive DEC value (Fig. 6a and b) because the TrIdx for these regions are significantly improved in response to temperature shift, but ClCon is not (Fig. 6a).

Combined, our analysis of CFTR TrIdx and ClCon relationships based on energetic insights derived from temperature shift reveals that the NBD1-ICL4 interface that includes F508del and residues 1061–1065 reflects the need for balance in stability of the fold for export versus plasticity for downstream function revealed by ClCon. While the interface provides a thermodynamic sensitive platform that contributes to both TrIdx and ClCon, there are many other regions such as residues 512–536 and the YKDAD di-acidic motif (Fig. 6a, b) that have differential energetic contributions to either ER export and/or downstream ClCon indicating a divergence in genome-encoded fold design reflecting the need for subcellular compartment specialization of sequence-to-function-to-structure relationships and their multigenerational evolution contributing to an individual's health and response to disease.

## Discussion

The mechanism by which the biological thermodynamic properties of the polypeptide sequence participate in folding trajectories in vivo to achieve function for any protein in response to different environmental conditions operating in unique cell types remains an enigma. This is particularly evident in eukaryotic cells where compartmentalization and membrane trafficking from and to specialized compartments provide a major advance to more precisely tune the fold for function[77].

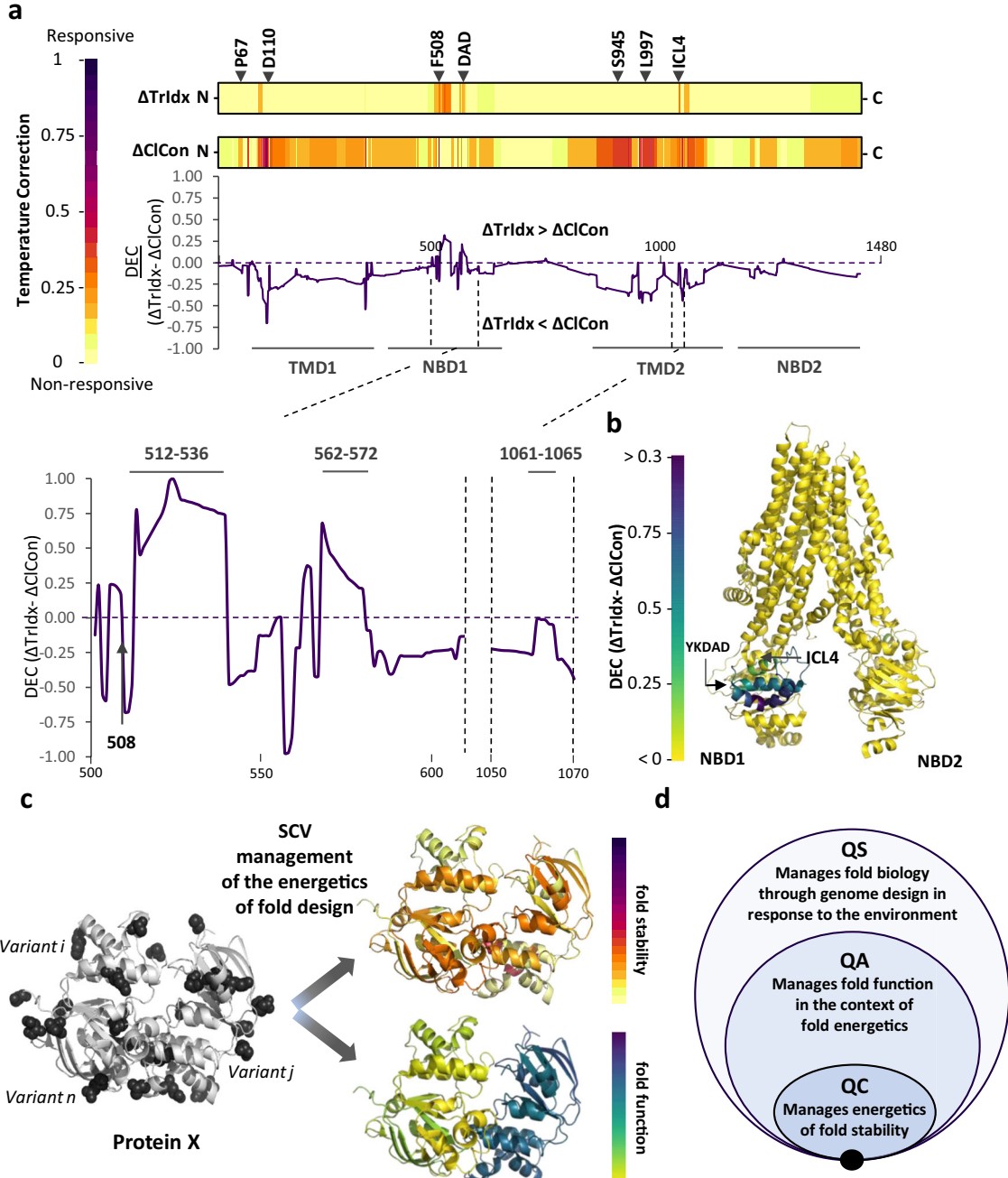

**Fig. 6 SCV code scoring global response to folding energetics across the entire CFTR fold. a** SCV-based linear code representation of the temperature correctability of CFTR TrIdx (ΔTrIdx, upper), ClCon (ΔClCon, middle) and the differential energetics based correction (DEC) plot displayed as the difference value (ΔTrIdx-ΔClCon) between the predicted ΔTrIdx and ΔClCon phenotype landscape values relative to that observed for WT values in response to temperature (Supplementary Data 2). **b** CFTR DEC functional structure highlighting regions such as the YKDAD motif that have a higher energetic contribution to the ER export than to the ClCon. **c** In general, combining the energetic features of the fold with functional assays using only a sparse collection of variants in the population and GPR-ML based VSP for any protein (generic example illustrated), we can determine the DEC for any residue relative to that observed for the WT protein as an SCV principled description of the physiological protein fold. The color scales reflect the differential response of the energetics of each residue to either stability or function which together contribute to the biology of the fold. **d** An SCV-based QS system dictating the biology of cell through genetic and epigenetic design of the genome in response to cell autonomous and non-autonomous signaling pathways. QS operates hierarchically with QC focused on the energetics of fold stability based on the physical-chemical properties of genome-encoded protein sequence that are independent of function. QA assesses the function of the fold throughout the cell in the context of proteostasis and physical-chemical environments unique to its compartmentalized organization, whereas QS encompasses the signaling pathways in genome design required to differentially manage biology in response to the environment. The proposed hierarchy of covariant relationships are defined by an ancient SCV principled language base built on biological non-equilibrium thermodynamic physical-chemical principles (black solid circle at base)[4].

Herein, we have applied the principle of SCV based on GPR to a sparse collection of variants found in the population to define how the biological thermodynamic properties of each residue contribute to function. We used CFTR as a prototypical example of the impact of variation on protein fold and function trajectories in response to the environment given the extensive number of variants contributing to disease in the CF population, but whose roles in disease remain largely mechanistically and clinically uncharacterized[27]. Export of F508del from the ER in response to reduced temperature is causally coupled to the co-translational program that, in response to the reduced rate of CFTR synthesis at 27 °C[46,78–80], has been shown to provide a biological thermodynamic advantage to its folding[34,71,78,81]. We have now extended this observation to every residue in the polypeptide chain. Our results are consistent with the evidence that reduced temperature energetically stabilizes the NBD1 domain in vitro[82–86]. We have now defined the covariant energetic relationships to show that a thermodynamic foundation for generating the CFTR fold in the ER is largely defined by the residues that contribute to the design of the NBD1-ICL4 interface facing the cytosol-stressing the integration of ER and cytosolic proteostasis pathways. Interestingly this interface is well conserved across different species (Supplementary Fig. 19). These key energetic folding events are coordinated allosterically with properties of the TM1/TM11 helices linked to ICL4 and ultimately, to more downstream domains in the coding sequence completing the assembly of the fold for function[86,87]. Our results are consistent with the fact that the folding of the NBD1-ICL4 region is the region primarily responsible for COPII engagement for trafficking from the ER via the di-acidic exit motif[21,22,63,88] and to prevent CFTR degradation by ERAD[52,53,64,66] and autophagy[68].

While the FoldX algorithm is well established in protein engineering and prediction of protein thermostability based on physical-chemical considerations[72–74], it failed to predict CFTR genotype to phenotype traits highlighting the fact that CFTR biogenesis leading to function in vivo is a multidimensional process that is not only defined by the genome encoded protein sequence but also by other parameters such as the translation speed, a wide range of protein interactions[64] and the activity of the proteostasis network[89–91] that manages protein folding dynamics. This observation is also consistent with the fact that CFTR trafficking and function require multiple conformational changes of flexible regions to couple folding to channel function. In general, SCV analyses focused on the global features of the protein fold suggests that theoretic thermostability predictions generated in vitro may have limited relevance to predict the dynamic biology of the functional fold in vivo, consistent with the now recognized highly dynamic and likely disordered states in protein design responsible for human biology in both health and disease[92,93].

While we showed that the fold required for the stability of the protein and export from the ER can contribute to stability at the PM (Supplementary Fig. 20), DEC codification reveals that a set of SCV-based biological thermodynamic relationships dictating export from the ER have no impact on the function at the cell surface. In contrast, a number of residues are exclusively used to manage ClCon at the cell surface. Thus, SCV demonstrates that ER export is not designed strictly, as currently assumed, to perform a quality control (QC) role related to acquisition of function. Instead, we propose the ER provides a more ancient and fundamental role in genome to proteome design based on SCV matrix principles[4], serving to construct a minimal core fold that balances the demands of nascent co-translational stability with the need for future plasticity to allow the cell to adapt function in response to the local environment. This view reflects the features controlling the dynamics of the fold inherent in the Anfinsen view[1] that "the protein fold'….'exists under conditions similar to those for which it was selected- the so-called physiological state"- in which it is the environment, not the snapshot[30] nor computed structure[94], that is the key to understanding the energetics based functionality of the folded state.

Given the genetic diversity captured by GnomAD (https://gnomad.broadinstitute.org/) and numerous other genome sequencing and phenotyping efforts (https://allofus.nih.gov/; https://www.ukbiobank.ac.uk/; https://www.ncbi.nlm.nih.gov/clinvar/), application of SCV principles through use of GPR machine learning can be tailored to any protein to independently assess stability or function, or any other feature using what we refer to as DEC codification that captures from a non-equilibrium thermodynamic perspective the impact of variation found in the population. DEC provides an approach to generate a more comprehensive and unified understanding from a mechanistic perspective the numerous genotypes contributing to health and disease[4,95,96] (Fig. 6c). Furthermore, since the GPR predicted functional value is based on the SCV principle that considers the covariance matrix information of a variant in the context of other variants, future development of the approach may enable the use of the SCV-based DEC codification to understand the complexity of heterozygosity (co-allelic) polymorphisms in rare disease, or the impact of variant heterogeneity in complex disease such as cancer. Therefore, combining sequence with function and structure through the application of GPR provides a fresh paradigm to assess biological responses revealed by variation in the population in the context of the SCV based matrix responsible for the protein fold in response to the local environment.

To reframe SCV based energetic principles responsible for fold management, we now hierarchically organize the cell as a 'quality system' (QS)[26] (Fig. 6d). At the most fundamental level is the concept of quality control (QC) which includes features contributing to the energetics of fold stability independent of function (Fig. 6d). Consistent with this view, the ER generates around one third of eukaryotic proteome without knowledge of downstream functionality. Many variants in rare diseases are efficiently exported from the ER that lack downstream function[4]. QC is based on unknown SCV relationships defined on a residue-by-residue basis in response to the energetics that dictate stability versus degradation that we have posited provides an evolutionary conserved SCV set-point for fold management[4]. We propose QC is only a subset of a more comprehensive quality assurance (QA) program that manages different energetic features of the fold required to achieve function, explaining why in vitro energetic calculations can fail to capture the biology of the fold (Fig. 6d). QA manages the function of the fold in response to energetics influenced by physiological state of the cell[1] (Fig. 6d) taking into account compartmentalized environments that we have referred to as SCV 'tolerance'[4]. QA is flexibly managed by proteostasis dependent folding and stress-responsive signaling pathways[89–91,97] including the unfolded protein response (UPR)[98,99], the integrated stress response (ISR)[100], and heat shock response (HSR)[101], among others. QS, defined by the genetic and epigenetic features of the whole-genome design of each cell type, manages globally the energetic tolerance of the fold in response to the environment required for host fitness (Fig. 6d) by balancing/rebalancing synthesis, stability, function, and degradation continuously throughout a lifespan[90,102]. We posit that SCV principled relationships managed by a QC-QA-QS system based on genomic variation in the population can reveal the role of diversity in driving the complex biology found in the population and the individual (Fig. 6d).

## Methods

**Cell culture and transfection.** HEK293 cells were cultured in Dulbecco's Modified Eagle Medium (DMEM), supplemented with 2 mM L-glutamine and 10% FBS at 37 °C (regular medium). Plasmid transfections were performed as followed.

Lipofectamine 3000 (Invitrogen), 100 ng of pBI-CMV2 empty vector or vector containing WT-CFTR or any of the 64 CFTR variants, were mixed with a ratio of 1/2/3 (DNA/Reagent/Lipofectamine) in OPTI-MEM. Optimization of the level of transfected DNA for WT and F508del was performed to define conditions for the optimal production CFTR in the cell (Supplementary Figs. 1–7). Using 96-well plates, 80% confluent HEK293 were transfected by adding the transfection mixture to a regular medium. Cells were incubated at 37 °C for 48 h and were then washed and incubated either at 37 °C or 27 °C for an additional 24 h in a regular medium.

**GFP fluorescence measurement**. Transfection efficiency was validated quantitatively using immunoblotting following transfection of HEK293 cells with 0, 12, 25, 50, 100 or 200 ng of pBI-CMV2-WT-CFTR or pBI-CMV2-F508del-CFTR after 24, 48 or 72 h (Supplementary Fig. 22) and by monitoring GFP fluorescence using a Synergy H1 hybrid reader (Biotek) (Supplementary Fig. 22b, c). We observe a similar level of GFP-fluorescence at a given time-point when HEK293 cells were transfected with the same quantity of either pBI-CMV2-WT-CFTR or pBI-CMV2-variant-CFTR. In all cases, the level of GFP- fluorescence and the level of CFTR generated by either the pBI-CMV2-WT-CFTR or pBI-CMV2-variant-CFTR using immunoblotting were directly comparable (Supplementary Fig. 22a–c). These data validate the use of GFP fluorescence as a metric to ensure uniform transfection efficiencies across the range of variants tested.

**Immunoblotting**. HEK293 cells lysates were prepared in 50 mM Tris-HCl, 150 mM NaCl, 1% Triton X-100 and 15 μg of total protein were separated using 7% SDS-PAGE. Proteins were transferred to nitrocellulose and probed with 3G11 (1.5 μg/ml) antibody for CFTR detection followed by a rat secondary antibody (1/10,000 dilution), and detection was performed by chemiluminescence. GFP fluorescence was monitored using Living Colors A.v. Monoclonal antibody (JL-8; Cat No. 632380; Clontech; 1/1000 dilution) followed by a rabbit secondary antibody (1/10,000 dilution) and detection performed using chemiluminescence. in order to have a proper visual reference for the migration of the band B and C glycoforms, we included a loading control of CFBE41o- -F508del cells cultured at reduced temperature.

**CFTR stability assay**. HEK293 cells were transfected with pBI-CMV2 empty vector or vectors containing WT-CFTR or any of the 64 CFTR variants tested and incubated for 48 h at 37 °C. The cells were subsequently incubated at 27 °C for 24 h. After 24 h at 27 °C, HEK293 cells transiently expressing CFTR were washed and incubated at 37 °C in regular medium supplemented with 50 μM of cycloheximide (CHX) final concentration for 0, 10, 30, 60, 120 min. CFTR variant stability was quantified by immunoblotting based on the difference between the measurements performed at 0 and 120 min of incubation post CHX addition. Due to the number of variants and conditions, only the first 4 h of cycloheximide treatment were monitored. While this short kinetic is not enough to monitor the entire degradation process of all variants, it was sufficient to distinguish between highly unstable variants such as the F508Del compared to more stable variants that still might have minor stability issues.

**Fluorescence membrane potential assay**. The FMP assay was performed as previously described[37]. HEK293 cells transiently expressing CFTR were washed twice in phosphate-buffered saline and incubated with the blue membrane potential dye dissolved in a modified PBS at 37 °C. The plates were read in a fluorescence plate reader at 37 °C following a dye equilibration period of 30 min. CFTR was stimulated with an activation cocktail (10 μM Forskolin and 50 μM genistein; final concentration) diluted in the same buffer. Fluorescence was monitored every 0.5 s for a total of 30 s. CFTR-mediated depolarization of the membrane was monitored as an increase in fluorescence and hyperpolarization or repolarization as a decrease. Of note, we observed that the impact of low temperature is, in general, more modest for variants which are processed and already have some TrIdx or ClCon activity compared to completely defective variants. This could be partially explained by the fact that the FMP assay as well as the Western blot and stability assays are semi-quantitative assays that are more efficient at monitoring improvement of non-trafficking/functional variants were the range for improvement is important and easily quantified.

**Adenoviral transduction**. 80% confluent -/-CFBE41o- -YFP (non-expressing CFTR cells) were transduced for 5 h in opti-Minimum Essential Medium (opti-MEM) in the presence of 10 μg/ml of polybrene (EMD Millipore Corp., Burlington, MA) with adenovirus carrying CFTR at a multiplicity of infection of 200. Then cells were washed with phosphate buffered saline (PBS) and fed with regular medium for 48 h prior to treatment.

**YFP quenching assay**. -/-CFBE41o- -YFP (non-expressing CFTR cells) were transduced with adenovirus carrying CFTR. Cells were then incubated at 37 °C or 27 °C for 24 h. Following treatment, cells were stimulated with a final concentration of 10 μm Fsk and 50 μm Gen for 15 min prior to the addition of PBS + NaI (replacement of NaCl with 137 mm NaI). Fluorescence was monitored every half second for a total of 30 s (3 s prior to the addition of NaI and 27 s after the addition

of NaI) using a Synergy H1 hybrid reader (BioTek, Winooski, VT). As a negative control, identical experiments were performed with -/-CFBE41o- -YFP cells not expressing CFTR. For each variant transduced in -/-CFBE41o- -YFP cells, their YFP fluorescence quenching relative to WT was measured by calculating the delta between the YFP fluorescence baseline (prior addition of PBS + NaI) and the YFP fluorescence after 27 s after addition of NaI normalized by the same ratio determined for -/-CFBE41o- -YFP cells transduced with adenovirus carrying WT CFTR.

**Variation spatial profiling**. The VSP analysis of the data was performed as previously described[4,6,7]. VSP uses a biological adaptation of geostatistics[39,40] based on a GPR-ML approach that provides a means to predict the distribution of a large range of known and unknown features using sparse sampling methods. GPR-based ML is an interpolation/regression method providing unbiased prediction based on the modeled SCV relationships generated by a molecular variogram. Below, we briefly describe how we apply classical GPR-ML through VSP to understand globally the role of variation in human disease. See refs. [1,4] for further details.

We first positioned the known variants by their sequence positions in the polypeptide chain on the $x$ axis coordinate and measurements of a biological parameter of these known variants on the $y$ axis coordinate to another biological parameter along the $z$ axis coordinate. For this purpose, suppose the $i^{th}$ (or $j^{th}$) observation in a dataset consists of a value $z_i$ (or $z_j$) at coordinates $x_i$ (or $x_j$) and $y_i$ (or $y_j$). The distance h between the $i^{th}$ and $j^{th}$ observation is calculated by:

$$h_{i,j} = \sqrt{((x_i - x_j)^2 + (y_i - y_j)^2)} \qquad (1)$$

The $\gamma(h)$-variance for a given distance ($h$) is defined by:

$$\gamma(h) = \frac{1}{2}(z_i - z_j)^2 \qquad (2)$$

where ($h$)-variance is the spatial variance (i.e., the degree of dissimilarity) of the $z$ value between the two observations, which is also the whole variance of $z$ value for one observation at the given separation distance h. The distance ($h$) and $\gamma(h)$-variance for all the data pairs are generated by the Eqs. (1) and (2). Then, the average values of $\gamma(h)$-variance for different distance intervals are calculated to plot $\gamma(h)$ versus h. A molecular variogram is generated from these calculations to define quantitatively the correlation between the spatial variance of $z$ changes and the separation distance defined by the $x$ and $y$ coordinates based on known variants. The distance where the model first flattens out is known as the range. Locations separated by distances closer than the range are spatially correlated, whereas locations farther apart than the range are not. The SCV at the distance ($h$) is calculated by C($h$) = C(0) − $\gamma(h)$, where C(0) is the covariance at zero distance representing the global variance of the data points under consideration (i.e., the plateau of the variogram). Based on the molecular variogram defined by the known variants we use a GPR-ML-based regression approach referred to as ordinary Kriging that has the least assumptions and is the most commonly used form to assign uncertainty to every unknown residue in the sequence[103]. Essentially, SCV can predict the unknown value by local weighted averaging the surrounding known values, where the weight associated with the known value is determined according to their positions both in relation to the unknown point and to one another. According to the values found in the molecular variogram, observations within close distance are usually highly correlated and have more weight for prediction. To solve the optimum and unbiased weights of SCV relationships, molecular geostatistics aims to minimize the variance associated with the prediction of the unknown value at location u, which is generated according to the expression:

$$\sigma_u^2 = E\left[(z_u^* - z_u)^2\right] = \sum_{i=1}^{n}\sum_{j=1}^{n}\omega_i\omega_j C_{i,j} - 2\sum_{i=1}^{n}\omega_i C_{i,u} + C_{u,u} \qquad (3)$$

where $z_u^*$ is the prediction value while $z_u$ is the true but unknown value, $C_{i,j}$ and $C_{i,u}$ are SCV between data points $i$ and $j$, and data points $i$ and $u$, respectively, and $C_{u,u}$ is the SCV within location $u$. $\omega_i$ is the weight for data point i. The SCV is obtained from the above molecular variogram analysis. To ensure an unbiased result, the sum of weight is set as one:

$$\sum_{i=1}^{n}\omega_i = 1 \qquad (4)$$

Equations (3) and (4) not only solved the set of weights associated with input observations, but also provided the minimized 'molecular variance' at location u which can be expressed as:

$$\sigma_u^2 = C_{u,u} - \left(\sum_{i=1}^{n}\omega_i C_{i,u} + \mu\right) \qquad (5)$$

where $C_{u,u}$ is the SCV within location $u$, $\omega_i$ is the weight for data point $i$, and $C_{i,u}$ are SCV between data points $i$ and $u$. $\mu$ is the Lagrange Parameter that is used to convert the constrained minimization problem in Eq. (3) into an unconstrained one. The resulting minimized molecular variance provides a weighted SCV score that represents the confidence for using the SCV relationships both within the input data points and in relation to the unknown locations. The confidence level is tightly linked with the distance range in the variogram. The shorter distance between the unknown point to the input data points, the higher confidence for using the SCV relationships for prediction. With the solved weights W, we can calculate the prediction of all unknown values to generate the complete phenotype

landscape by the equation:

$$z_u^* = \sum_{i=1}^{n} \omega_i z_i \qquad (6)$$

where $z_u^*$ is the prediction value for the unknown data point $u$, $\omega_i$ is the weight for the known data point, and $z_i$ is the measured value for data point $i$.

In the final prediction, for each unknown position, minimal 5 nearby variants and maximal 15 nearby variants were included in the weighted averaging calculation (i.e., Eq. (6)). All the procedures were performed in the ordinary Kriging module in GS+, version 10 from gammadesign software (https://geostatistics.com/).

To map the function predictions onto structure, we assign the prediction value with highest confidence to each residue to generate a functional structure that illustrates all values interpolated from the sparse collection variants used to generate the phenotype landscape. PDB:5UAK is used for CFTR structure mapping[30]. All the structural presentations were produced by the software of PyMOL.

**Statistics and reproducibility**. The statistical validation methods to assess the performance of the VSP strategy used in this study include a leave-one-out cross-validation and a validation by an external dataset when possible. The default validation method is leave-one-out cross-validation because of the small sample size modeling. In the leave-one-out cross-validation, all data are initially used to build the molecular variogram and geostatistical models. We remove each data point, one at a time and use the rest of the data points to predict the missing value. We repeat the prediction for all data points and compare the prediction results to the measured value to generate the Pearson's PCC-value and its associated $p$ value (ANOVA test, with null hypothesis with the coefficient equal to zero).

For the external dataset validation of ClCon and TrIdx predictions at 37 °C (Supplementary Fig. 6), we considered the results of 21 CF variants from a separate study[33,43] that were not used for training. Predicted $z$ values were generated by feeding the model with $x$- and $y$- values, and subsequently compared to the observed values by Pearson's correlation analysis and $p$ value calculation (ANOVA test).

SCV-based functional insights are able to achieve a significantly more accurate prediction to inform on unappreciated genotype to phenotype relationships as described[4]. These relationships cannot be defined using other predictive model such as PolyPhen-2, SIFT, CADD, M-CAP, Envision[104], Grantham, as well as POSE[28] that were trained with the same dataset[4].

**Reporting summary**. Further information on research design is available in the Nature Research Reporting Summary linked to this article.

## Data availability

Atablell trafficking and functional data for each CFTR variants described in this project can be found in the Supplementary Information (Supplementary Figs. 1–7, 9–14). All uncropted and unedited blots related to Figs. 3a, 4a, Supplementary Figs. 1–7, 16 are available in Supplementary Figs. 23–94. In addition, all source data presented in the main figures are available in the supplementary Data 3.

## Code availability

A demo input data, the R-codes for generating phenotype landscape and instructions were previously published[3] and are available as a Zip file named "ExampleCode.Zip", which is uploaded to https://doi.org/10.17632/ycw667nv5f.1.

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

## Acknowledgements

Grant support provided by NIH HL095524, DK051870, AG070209, AG049665, HL141810, and HG010881 to W.E.B. F.A. and C.W. are past recipients of CFFT Post-doctoral Fellowships. We thank P. Thomas (UT Southwestern Medical School) for providing vectors encoding CFTR2 variants.

## Author contributions

F.A. designed and performed the experiments; F.A., C.W., and W.E.B. contributed to concepts and wrote the manuscript.

## Competing interests

The authors declare no competing interests.
