## [Peer Review File · Communications Biology]

Reviewers' comments:

Reviewer #1 (Remarks to the Author):

Reviewer Information:

Understanding the layers in Protein quality control and the fine balance between propensities leading to proteostasis and proteosomal degradation has been an enigma in protein biophysics. The current article has touched upon Protein Quality Assurance considering biological thermodynamic properties of polypeptide chain and aims to bridge the sequence-function paradigm.

The article is generally well planned and discusses the interesting correlation between genome variation and the thermodynamic properties of the protein. The authors have considered CFTR as the model system owing to its implication in the clinical manifestation of CF. Authors have used Gaussian process regression-based principle (SCV) to establish the correlation and understand how certain forms CFTR are particularly more toxic from a clinical perspective. Use of SCV has been reported to be particularly useful in assigning biological thermodynamic features to the protein primary structure on a per-residue basis. This has helped demystifying the role of residues constituting CFTR polypeptide sequence in cell compartment specific fates viz. stability and/or transport/function. Authors have demonstrated a "thermodynamically sensitive region of the CFTR fold" which regulates the energetics of ER transport. This is an interesting observation and appears to be a promising approach which could be of general value to the studies involving PQC (Protein quality control) and Protein quality assurance in the contexts of proteinopathy and/or altered cellular protein abundance. Further SCV could have an interesting and more broad application in understanding how and why certain point mutations confined in certain stretches of a protein leads to higher propensities in mis-folding and concomitant progression to the aggregation pathway. The article presents a potential direction in area of Protein Quality Control/Assurance and Protein biophysics in general with potential application for other candidate proteins. I recommend the article to be considered for publication but I do have some general comments and suggestion, which if incorporated would appeal to even a larger readership.

1. The authors have used the interesting combination of trafficking index (TrIdx) and conductance measurements (CICon) [Line 120 to 125] to catalogue the transport efficiency and functionality. Since this is an important study and for building the message of the paper, I would recommend a more detailed and lucid explanation of what these two measurements are.
2. Line 285 to 288 mentions application of VSP to generate a temperature sensitive (TS) delta (Δ) (Stride)-phenotype landscape to understand the TS response on traffic index (TrIdx). This section is hard to comprehend and would be a better read if made smooth with better explanation.
3. The results section as such is a little too much with information and sometimes hard to follow with occasional disruption to the main message. It would be very helpful for the readers if the results section is made more organized and smoother with lucid explanation for the main text figures.
4. Line 311 to 316 discusses "thermodynamic sensitive SCV relationships that capture the energetic properties across the entire CFTR nascent fold in response to temperature shift in vivo reveals for the first time an integrated thermodynamic sensitive region comprising the NBD1-ICL4 interface that dictates SCV tolerance (Wang and Balch, 2018) of the fold for either recognition by the COPII vesicle machinery for export". Have the authors attempted in generating all residue mutational tolerance profile and seeing how residues constituting NBD1-ICL4 interface shows up in the tolerance matrix and which positions are more constrained with very low tolerance. It could be out of the scope for the main text of the article but authors could consider this analysis and if interesting results show up, could integrate with the main story.
5. It would have been interesting if authors had included some evolutionary co-variation analysis utilising the evolutionarily sampled sequence space of CFTR. The approach presented in the paper is interesting and makes an evolutionary investigation more obvious. If it is beyond the current scope or a part of a follow up story, authors should at-least discuss evolutionary perspectives in the discussion.
6. The approach stands interesting and potentially useful to explore the QC layers in proteins more universally. Can the authors speculate on the potential of the approach presented for more complex proteins viz. metallo-proteins where the folding events and follow-up stability/degradation

balance is multi layered and tightly orchestrated by both polypeptide sequence information as well as local stabilities inside the metal pockets. It would be interesting if this angle is considered and discussed when SCV's potential universality is being discussed.

7. MINOR: The article has grammatical and spelling mistakes at places which have to be taken care while working on the revised draft.

Reviewer #2 (Remarks to the Author):

Summary of the work

In this work, the authors use the principle of spatial covariance to predict and study the thermodynamic properties of each residue of CFTR in relation to two major processes in the cell: ER export and channel conductance at the plasma membrane. In order to do this, they individually express 64 disease-relevant CFTR missense variants (+ WT) in HEK293 cells, and measure the CFTR trafficking index (TrIdx) (based on the fraction of CFTR migrating as the Golgi-processed vs ER-retained glycoform on western blot) and chloride ion conductance (ClCon) (based on a membrane potential responsive fluorescent dye). Thermodynamic information is obtained by performing these same measurements at two different temperatures (physiological 37°C, and 27°C), and by halting translation (with cycloheximide), so can protein stability. Combining these various metrics with the protein sequence position into 3D space and analyzing the spatial relationships of the known datapoints (mutations), a predictive model can be built from which the full spectrum of residues across the polypeptide sequence can be scrutinized in relation to the chosen metrics.

This study thus dissects the contributions of single amino acid residues along the CFTR polypeptide chain to ER export, post-ER stability, channel function at the plasma membrane, and their associated thermodynamic sensitivities. Two major conclusions from this work are a) the identification of a specific thermodynamically responsive region in the fold that determines ER export, and b) that certain residues or regions of the channel are important for channel fold and function beyond and independent of ER processes. The latter is contrary to the somewhat widespread notion that most detrimental CFTR mutations affect protein function solely through misfolding in the ER and engagement of the ER QC machinery or interference with ER export. The observation that the effect of many of these mutations does not match up with previous thermodynamic measurements done in vitro emphasizes the importance of the cellular environment in determining energetics of the fold for function.

The authors frame these different pre- and post-ER residue contributions into the broader concept of 'quality assurance', which they define as influencing function of the fold due to changing physiological state of the cell or cellular compartment.

Overall Impression

This is a well thought-out original study and a convincing example of the power of SCV for biologically-relevant discovery. The authors chose a highly relevant and substantial yet broad set of mutations to build the landscapes, the biological assays are straightforward, the controls and comparisons with existing datasets and tools are appropriate, and the basic statistics are sound. The figures are well-designed and clear, all data is present and well-organized in the supplementary, and the availability of demo R code is appreciated. The technical details of VSP analysis are beyond my expertise to be able to comment on their accuracy, but the principle of SCV is well-explained in the main text.

The conclusions are powerful and really interesting both for the CFTR field, where large numbers of disease-associated variants remain mostly uncharacterized, as well as anyone studying the fundamentals of protein folding and the cellular processes affecting these, since it highlights the influence of cellular compartment and physiological state of the cell on fold energetics.

The one caveat I would note is that the authors promote SCV as a great tool to study fold-function relationships and this has indeed worked out well in this case, but the limitations of the method for proteins other than CFTR are not discussed at all. As with many proteins, it seems that the

individual success rate highly depends on the design of ingenious assays that report on that specific protein's function.

Specific Comments

Minor comments:

1. Line 19 (abstract): "...we globally assign residue-by-residue the biological thermodynamic properties..."
2. Line 22 (abstract): "... that contributes to its endoplasmic reticulum (ER) export"
3. Line 98: "to address the role of energetics..."
4. Line 128: "from other mutation-based modeling approaches"
5. Line 306: "to confer COPII-independent export from the ER"
6. Line 356: "To assess the impact of energetics on ClCon, cells expressing the different variants were incubated..."
7. Lines 366 and 368: statistical test used?
8. Line 421: Fig S6A not S8A
9. Line 710 (Fig 1 legend): first use of ClCon when reading the manuscript figure legend so write it in full: "chloride conductance (ClCon)"
10. In the blots in Fig 3A and Fig S1, what is the reference sample exactly? This isn't mentioned in the methods or in the figure legends.
11. Also in the blots in Fig 3A and Fig S1, I'm assuming the two lanes per condition are biological replicates? Perhaps also good to confirm this somewhere.
12. Line 748 (Fig 3 legend): on the pdf the 27°C data shows up as purple not blue?
13. Fig S3 is missing a legend for the colors used.
14. Line 905: Fig S9 instead of S7.
15. This is somewhat subjective and up to the authors' discretion, but I do not find the title of the manuscript clear nor engaging for the reader. My recommendation is to mention CFTR since this is the main subject of the work, to emphasize that the thermodynamics measurements were done in vivo, and perhaps to strip the reference to quality assurance since this is more of a conceptual framework rather than an objective observation. One example could be "Spatial Covariance analysis reveals the residue-by-residue thermodynamic contribution to CFTR fold and function in the cell"

Reviewer #3 (Remarks to the Author):

In this study, Anglès et al define a novel approach for folding quality assessment applied to CFTR, the protein which when mutated causes cystic fibrosis. The authors use 64 disease-causing variants and assess their impact on processing/trafficking (defined by a common parameter, here recoined as TrIdx) and function (defined by the chloride conductance ClCon, assessed through an indirect approach). Based on these two parameters and on how the variants respond to low temperature, the authors define several "landscapes" that can be used to define the impact of every residue in the fold of CFTR.

This is an innovative approach, and the results are of significant impact for the field. However, some aspects need to be clarified so that the interpretation of results is fully supported by the data presented.

Major concerns

1. Although the authors define this approach in a broad sense, the manuscript is in fact about CFTR folding and how it impacts trafficking and processing – the title should reflect this.
2. CFTR is normally expressed in epithelial cells. In this study, the authors use mainly HEK cells and in some cases also HeLa and FRT cells – and the results are comparable. All the results are obtained with overexpression of CFTR. At one point, the authors comment that the results presented match those presented elsewhere in human bronchial cells (line 152-154, ref Han et al, 2018). Considering the relevance of the results presented, it is strongly suggested that the authors perform validation with CFTR expressed in bronchial epithelial cells – the CFBE cell model (although also based in overexpression) is widely accepted. The variants chosen for this validation should include the different "patterns" described in the manuscript.
3. The TrIdx should be properly described in the text and the normalizations performed clearly

stated in the legends.

4. The authors should comment on the fact that the “good” correlations observed are greatly dependent on the variants which exhibit no function – see e.g. Fig.S4 and the high numbers of variants clustered around the origin of the plot.

5. The results observed for N1303K-CFTR are quite puzzling. This mutation is relatively common (the second most common trafficking mutation). Its effect upon CFTR processing is dramatic – there are many reports in the literature showing this. However, in the current study, N1303K-CFTR is processed and the mature form is as stable as that of WT-CFTR. This is probably the weakest point in the manuscript that threatens the “relevance” of the study as a whole – if such a well known mutation does not match here what is known, how can the predictions for less known mutations be considered trustable?

6. The same applies to other mutations for which data seems to be in conflict with previous descriptions – e.g. R560T previously shown not to respond to low temperature, I1234V which is in fact mainly a splicing mutation...

7. The impact of low temperature seems to be much more modest for variants which are processed and already have some function – the authors need to discuss how the sensitivity of the assays used can impact the overall interpretation of data. Naturally, the scope for improvement is much smaller in cases for which there is function.

Minor concerns

8. Please comment to what does the number 1,000 disease-causing variants (l.101) refer to. A total of over 2,000 variants has been reported, with some 350 characterized as disease-causing.

9. HeLa and FRT cells seem to be missing from the “Materials and Methods” section.

10. Check all legends to make sure that everything is properly mentioned – e.g. in Fig.S3, the blue/orange are not explained.

Reviewers' comments and Response (where necessary indicated in blue and underlined in the main text):

Reviewer #1 (Remarks to the Author):

Reviewer Information. Understanding the layers in Protein quality control and the fine balance between propensities leading to proteostasis and proteasomal degradation has been an enigma in protein biophysics. The current article has touched upon Protein Quality Assurance considering biological thermodynamic properties of polypeptide chain and aims to bridge the sequence-function paradigm. The article is generally well planned and discusses the interesting correlation between genome variation and the thermodynamic properties of the protein. The authors have considered CFTR as the model system owing to its implication in the clinical manifestation of CF. Authors have used Gaussian process regression-based principle (SCV) to establish the correlation and understand how certain forms CFTR are particularly more toxic from a clinical perspective. Use of SCV has been reported to be particularly useful in assigning biological thermodynamic features to the protein primary structure on a per-residue basis. This has helped demystifying the role of residues constituting CFTR polypeptide sequence in cell compartment specific fates viz. stability and/or transport/function. Authors have demonstrated a “thermodynamically sensitive region of the CFTR fold” which regulates the energetics of ER transport. This is an interesting observation and appears to be a promising approach which could be of general value to the studies involving PQC (Protein quality control) and Protein quality assurance in the contexts of proteinopathy and/or altered cellular protein abundance. Further SCV could have an interesting and more broad application in understanding how and why certain point mutations confined in certain stretches of a protein leads to higher propensities in mis-folding and concomitant progression to the aggregation pathway. The article presents a potential direction in area of Protein Quality Control/Assurance and Protein biophysics in general with potential application for other candidate proteins. I recommend the article to be considered for publication but I do have some general comments and suggestion, which if incorporated would appeal to even a larger readership.

1. The authors have used the interesting combination of trafficking index (TrIdx) and conductance measurements (ClCon) [Line 120 to 125] to catalogue the transport efficiency and functionality. Since this is an important study and for building the message of the paper, I would recommend a more detailed and lucid explanation of what these two measurements are.

We have now provided additional details regarding TrIdx (Line 138-139) and ClCon (149-151) measurements.

2. Line 285 to 288 mentions application of VSP to generate a temperature sensitive (TS) delta (Δ) (Stride)-phenotype landscape to understand the TS response on traffic index (TrIdx). This section is hard to comprehend and would be a better read if made smooth with better explanation.

We have now re-written the section (Line 297-304) to facilitate comprehension.

3. The results section as such is a little too much with information and sometimes hard to follow with occasional disruption to the main message. It would be very helpful for the readers if the results section is made more organized and smoother with lucid explanation for the main text figures.

We have now modified the Results section to facilitate comprehension.

4. Line 311 to 316 discusses “thermodynamic sensitive SCV relationships that capture the energetic properties across the entire CFTR nascent fold in response to temperature shift in vivo reveals for the first time an integrated thermodynamic sensitive region comprising the NBD1-ICL4 interface that dictates SCV tolerance (Wang and Balch, 2018) of the fold for either recognition by the COPII vesicle machinery for export”. Have the authors attempted in generating all residue mutational tolerance profile and seeing how residues constituting NBD1-ICL4 interface shows up in the tolerance matrix and which positions are more constrained with very low tolerance. It could be out of the scope for the main text of the article but authors could consider this analysis and if interesting results show up, could integrate with the main story.

We have provided a Table with all residue’s mutational thermodynamic tolerance profile (Table S2) and have now describe more in detail which positions are more constrained with very low variability in the ICL4 region (Line 320-322).

5. It would have been interesting if authors had included some evolutionary co-variation analysis utilizing the evolutionarily sampled sequence space of CFTR. The approach presented in the paper is interesting and makes an evolutionary investigation more obvious. If it is beyond the current scope or a part of a follow up story, authors should at-least discuss evolutionary perspectives in the discussion.

We performed the sequence alignment of CFTR from different species, such as human, bovine, rat, mouse and African clawed frog (Fig. S8). The three sequence fragments important for CFTR thermodynamics that are required in ER trafficking, i.e., F508, DAD di-acidic ER export code, and ICL4 (Fig. 3B-D), are conserved across different species (Fig. S8). This result suggests that the thermodynamic essential regions that we identified through GPR are evolutionarily critical for the CFTR fold.

6. The approach stands interesting and potentially useful to explore the QC layers in proteins more universally. Can the authors speculate on the potential of the approach presented for more complex proteins viz. metallo-proteins where the folding events and follow-up stability/degradation balance is multi layered and tightly orchestrated by both polypeptide sequence information as well as local stabilities inside the metal pockets. It would be interesting if this angle is considered and discussed when SCV’s potential universality is being discussed.

We do have a study currently under peer review that applies SCV to understand how the structural thermodynamic stability of SARS-CoV-2 polymerase impacts the COVID-19 pandemic behavior. In this study, we found a metal pocket presents a SCV thermodynamic core that SARS-CoV-2 optimizes through variation to improve its fitness.

7. *Minor*: The article has grammatical and spelling mistakes at places which have to be taken care while working on the revised draft.

Reviewer #2 (Remarks to the Author):

Summary of the work. In this work, the authors use the principle of spatial covariance to predict and study the thermodynamic properties of each residue of CFTR in relation to two major processes in the cell: ER export and channel conductance at the plasma membrane. In order to do this, they individually express 64 disease-relevant CFTR missense variants (+ WT) in HEK293 cells, and measure the CFTR trafficking index (TrIdx) (based on the fraction of CFTR migrating as the Golgi-processed vs ER-retained glycoform on western blot) and chloride ion conductance (ClCon) (based on a membrane potential responsive fluorescent dye). Thermodynamic information is obtained by performing these same measurements at two different temperatures (physiological 37°C, and 27°C), and by halting translation (with cycloheximide (CHX)), so can protein stability. Combining these various metrics with the protein sequence position into 3D space and analyzing the spatial relationships of the known datapoints (mutations),

a predictive model can be built from which the full spectrum of residues across the polypeptide sequence can be scrutinized in relation to the chosen metrics. This study thus dissects the contributions of single amino acid residues along the CFTR polypeptide chain to ER export, post-ER stability, channel function at the plasma membrane, and their associated thermodynamic sensitivities. Two major conclusions from this work are a) the identification of a specific thermodynamically responsive region in the fold that determines ER export, and b) that certain residues or regions of the channel are important for channel fold and function beyond and independent of ER processes. The latter is contrary to the somewhat widespread notion that most detrimental CFTR mutations affect protein function solely through misfolding in the ER and engagement of the ER QC machinery or interference with ER export. The observation that the effect of many of these mutations does not match up with previous thermodynamic measurements done in vitro emphasizes the importance of the cellular environment in determining energetics of the fold for function. The authors frame these different pre- and post-ER residue contributions into the broader concept of ‘quality assurance’, which they define as influencing function of the fold due to changing physiological state of the cell or cellular compartment.

Overall Impression. This is a well thought-out original study and a convincing example of the power of SCV for biologically-relevant discovery. The authors chose a highly relevant and substantial yet broad set of mutations to build the landscapes, the biological assays are straightforward, the controls and comparisons with existing datasets and tools are appropriate, and the basic statistics are sound. The figures are well-designed and clear, all data is present and well-organized in the supplementary, and the availability of demo R code is appreciated. The technical details of VSP analysis are beyond my expertise to be able to comment on their accuracy, but the principle of SCV is well-explained in the main text.

The conclusions are powerful and really interesting both for the CFTR field, where large numbers of disease-associated variants remain mostly uncharacterized, as well as anyone studying the fundamentals of protein folding and the cellular processes affecting these, since it highlights the influence of cellular compartment and physiological state of the cell on fold energetics.

The one caveat I would note is that the authors promote SCV as a great tool to study fold-function relationships and this has indeed worked out well in this case, but the limitations of the method for proteins other than CFTR are not discussed at all. As with many proteins, it seems that the individual success rate highly depends on the design of ingenious assays that report on that specific protein’s function.

Specific Comments

Minor comments:

1. Line 19 (abstract): “..we globally assign residue-by-residue the biological thermodynamic properties...”
2. Line 22 (abstract): “.. that contributes to **its** endoplasmic reticulum (ER) export”
3. Line 98: “to address the role **of** energetics...”
4. Line 128: “from other mutation-based modeling approaches”
5. Line 306: “to confer COPII-independent export **from** the ER”
6. Line 356: “To assess **the** impact of energetics on ClCon, **cells expressing the different** variants were incubated...”

We thank the reviewer for taking the time to precisely highlights all grammatical and spelling mistakes. We have now corrected the manuscript accordingly.

7. Lines 366 and 368: statistical test used?

A two-tailed Students’ t-test was used for the statistical test. We indicated the statistical test method in the legend of **Fig.S10**. Specifically, the spatial variance of ClCon were computed for all the possible variant pairs (in total 2016 pairs) at 37°C or 27°C. Then the spatial variance values of ClCon for all 2016 pairs were compared between 37°C and 27°C by using a two-tailed students’ t-test. The corresponding p-value was reported ($p = 0.04$). Similarly, the distances based on sequence position (VarSeq) and trafficking index (TrIdx) were computed for all 2016 variant pairs at 37°C or 27°C, and then compared with each other by using a two-tailed students’ t-test. The corresponding p-value was reported ($p = 5 \times 10^{-6}$).

8. Line 421: Fig S6A not S8A

9. Line 710 (Fig 1 legend): first use of ClCon when reading the manuscript figure legend so write it in full: “chloride conductance (ClCon)”

We have written in full ClCon and TrIdx when first used in the figure legend.

10. In the blots in Fig 3A and Fig S1, what is the reference sample exactly? This isn’t mentioned in the methods or in the figure legends.

WT and F508Del CFTR at 37°C have nearly no band B or C, respectively. In order to have a proper visual reference for the migration of the band B and C glycoforms, we usually include a loading control of

CFBE410- -F508del cells cultured at reduced temperature. We have now described this loading control in the Methods Section (Line 620-621).

11. Also in the blots in Fig 3A and Fig S1, I'm assuming the two lanes per condition are biological replicates? Perhaps also good to confirm this somewhere.

We have now updated the two legends accordingly.

12. Line 748 (Fig 3 legend): on the pdf the 27°C data shows up as purple not blue?

We change the legend accordingly.

13. Fig S3 is missing a legend for the colors used.

We updated the legend accordingly.

14. Line 905: Fig **S9** instead of S7.

We corrected the labels of the figure in the main text.

15. This is somewhat subjective and up to the authors' discretion, but I do not find the title of the manuscript clear nor engaging for the reader. My recommendation is to mention CFTR since this is the main subject of the work, to emphasize that the thermodynamics measurements were done in vivo, and perhaps to strip the reference to quality assurance since this is more of a conceptual framework rather than an objective observation. One example could be "Spatial Covariance analysis reveals the residue-by-residue thermodynamic contribution to CFTR fold and function in the cell"

To reflect the manuscript focus on CFTR, we have now, change the title to: "Spatial Covariance Reveals the Residue-by-Residue Thermodynamic Contribution to CFTR Fold and Function in the Cell"

Reviewer #3

(Remarks to the Author): In this study, Anglès et al define a novel approach for folding quality assessment applied to CFTR, the protein which when mutated causes cystic fibrosis. The authors use 64 disease-causing variants and assess their impact on processing/trafficking (defined by a common parameter, here coined as TrIdx) and function (defined by the chloride conductance ClCon, assessed through an indirect approach). Based on these two parameters and on how the variants respond to low temperature, the authors define several “landscapes” that can be used to define the impact of every residue in the fold of CFTR. This is an innovative approach, and the results are of significant impact for the field. However, some aspects need to be clarified so that the interpretation of results is fully supported by the data presented.

Major concerns:

1. Although the authors define this approach in a broad sense, the manuscript is in fact about CFTR folding and how it impacts trafficking and processing – the title should reflect this.

To reflect the manuscript focus on CFTR, we have now, as recommended by another reviewer as well, change the title to: "Spatial Covariance Reveals the Residue-by-Residue Thermodynamic Contribution to CFTR Fold and Function in the Cell"

2. CFTR is normally expressed in epithelial cells. In this study, the authors use mainly HEK cells and in some cases also HeLa and FRT cells – and the results are comparable. All the results are obtained with overexpression of CFTR. At one point, the authors comment that the results presented match those presented elsewhere in human bronchial cells (line 152-154, ref Han et al, 2018). Considering the relevance of the results presented, it is strongly suggested that the authors perform validation with CFTR expressed in bronchial epithelial cells – the CFBE cell model (although also based in overexpression) is widely accepted. The variants chosen for this validation should include the different “patterns” described in the manuscript.

We did not express any variants in HBE cells. We did not use HBE cells to perform our thermodynamic screen on the CFTR variant library as these cells are hard to transfect (less than 20% plasmid transfection efficiency) and it would have been difficult to detect any significant signal above background. Therefore, we decided to perform the experiments in HEK cells (more than 95% transfection efficiency). However, we have a small set of 10 CFTR variants + CFTR WT in adenoviruses. We have now transduced in HBE cells these variants and have now provided additional data in HBE cells (Fig. S5) that validate our observation in HEK cells (Line 158-162). For example, F508Del TrIdx is also highly improved in HBE cells while the S492F is still mostly not responding (Fig. S5) (Line 291-292 & 379).

3. The TrIdx should be properly described in the text and the normalizations performed clearly stated in the legends.

We have now provided an additional description of the TrIdx and its normalization (Line138-139). This description was added to the figure legends.

4. The authors should comment on the fact that the “good” correlations observed are greatly dependent on the variants which exhibit no function – see e.g., Fig.S4 and the high numbers of variants clustered around the origin of the plot.

To ensure that the “good” correlation between the normalized ClCon in HEK 293 and FRT cells is not entirely dependent on the variants which exhibit no function, we added a second comparison (**Fig.S4 right panel**). We removed from the global comparison (**Fig.S4 left panel**) all FRT variants previously published with less than 5% function (normalized to WT) leaving 27 variants. The comparison of these 27 variants yielded a PCC score of 0.68. Removing more than half of the data point did not drastically changed the PCC showing that the removed variants do not greatly bias the correlation.

5. The results observed for N1303K-CFTR are quite puzzling. This mutation is relatively common (the second most common trafficking mutation). Its effect upon CFTR processing is dramatic – there are many reports in the literature showing this. However, in the current study, N1303K-CFTR is processed and the mature form is as stable as that of WT-CFTR. This is probably the weakest point in the manuscript that threatens the “relevance” of the study as a whole – if such a well-known mutation does not match here what is known, how can the predictions for less known mutations be considered trustable?

As reported in our study, N1303K band C was previously observed, and no significant improvement of Band C was observed after temperature shift (see Rapino et al., 2015, Plos One). However, the function of N1303K has been shown to be partially restored by cell culture at low temperature (He et al., 2021, MBoC) independently of Band C restoration. Interestingly, while N1303K was reported to be less stable than WT, no significant destabilization of N1303K was observed after the first 4h of treatment with CHX (Rapino et al., 2015, Plos One). In general, all of our data regarding N1303K is in perfect agreement we previously published data.

The corrected F508Del band C is mostly degraded after 4h of CHX treatment, which we used as a reference when starting these experiments. Due to the number of variants and conditions, we restricted ourselves to the first 4 hours of CHX treatment. While this short kinetic time-frame is not enough to monitor the entire degradation process of all variants, it was already sufficient to distinguish between highly unstable variants such as the F508Del and more stable variants (that might still display some stability issues). We acknowledge the limitation of this experimental procedure and have added comments in the material and method section to address this concern (Line 628-632).

6. The same applies to other mutations for which data seems to be in conflict with previous descriptions – e.g., R560T previously shown not to respond to low temperature, I1234V which is in fact mainly a splicing mutation...

Regarding R560T, there were already conflicting views on its behavior as some investigators see a small amount of band C (Lopes-Pacheco et al., 2016, Chembiochem) while other do not (Farihna et al., 2013, Chem. Biol.). While the latter also observed no band C after 27°C incubation (Farihna et al., 2013 Chem. Biol.) others see an improvement of band C (4-fold change relative 37°C) after 27°C incubation (Lopes-Pacheco et al., 2016, Chembiochem). Our study shows only a modest 20% improvement in TrIdx for this variants which is not in conflict with some of the literature. The result may reflect the cell type used, the laboratory specific experimental conditions, age of cells in tissue culture, etc. I1234V trafficking and

activity previously published are similar to that of WT (Phuan et al., 2020, JCF) which is in line with what was observed in our lab. Regarding the stability of this variant, while it is quite stable compared to strongly defective variants, on the Western blot this variant is a bit less stable than others in the same region such as the S1251N following CHX treatment. However, this impact on the stability does not seem to impact its function at the plasma membrane in our transient overexpressing system. While it is known that the I1234V mutation is mostly detrimental due to its impact on RNA splicing we did not find any data in the literature showing the variant stability compared to WT. In general, to the best of our knowledge nothing in the literature is in conflict with the observation that I1234V may have a modest impact on CFTR stability.

7. The impact of low temperature seems to be much more modest for variants which are processed and already have some function – the authors need to discuss how the sensitivity of the assays used can impact the overall interpretation of data. Naturally, the scope for improvement is much smaller in cases for which there is function.

We agree that the Western blot and FMP assay are semi-quantitative assays that are more efficient at monitoring improvement of non-trafficking/functional variants where the extent for improvement is easily observed. For example, it is hard to quantify with accuracy a small modulation of band C on a Western blot when the variants is already behaving like WT. We have now provided comments on how the sensitivity of the assays used can impact the overall interpretation of data (Line 641-646).

Minor concerns

8. Please comment to what does the number 1,000 disease-causing variants (l.101) refer to. A total of over 2,000 variants has been reported, with some 350 characterized as disease-causing.

We apologize for the confusion, we meant around 2,000 variants reported in the population with approximately 50% predicted missense mutation (to date, 2380 mutations observed in the population with 1066 predicted missense mutations). We re-phrased it in the manuscript (Line 101-102).

9. HeLa and FRT cells seem to be missing from the “Materials and Methods” section.

The HeLa and FRT data sets used for the different correlations with our HEK data set were previously published (Sosnay et al., 2013, Nat. Gen.) and were referenced in the manuscript when used (Line 142,153, 260, 860, 883 & 884)

10. Check all legends to make sure that everything is properly mentioned – e.g., in Fig.S3, the blue/orange are not explained.

We have now updated all legends.

REVIEWERS' COMMENTS:

Reviewer #1 (Remarks to the Author):

The authors have carefully addressed the points that I had raised and have revised the manuscript accordingly.

Reviewer #2 (Remarks to the Author):

My comments have been sufficiently addressed and I approve of the revised version of the manuscript.

Reviewer #3 (Remarks to the Author):

The authors introduced significantly changes in the manuscript and most of my concerns were properly addressed. There are however still a few major concerns that are mainly highlighting some bias or lack of rationale in the way the subjects are addressed. They should be solved – or, if no rationale explanations are proposed, simply removed from the manuscript.

Previous comment #5

The results with N1303K remain the weakest aspect of the work presented here. Although there has been at least one report showing a band of higher molecular mass than band C in cells expressing N1303K-CFTR, there are dozens of papers in which N1303K-CFTR is not processed. At least a comment on this controversial aspect needs to be added. The results are in fact so puzzling that doubt remains if the construct is really bearing N1303K.

Previous comment #6

I1234V has been shown to affect splicing – thus decreasing considerably the amount of full length CFTR (bearing I1234V) that is synthesized. It doesn't make sense to treat it as a simple missense mutation – and so this mutation should be removed.

Reviewer 3 Concerns:

The authors introduced significantly changes in the manuscript and most of my concerns were properly addressed. There are however still a few major concerns that are mainly highlighting some bias or lack of rationale in the way the subjects are addressed. They should be solved – or, if no rationale explanations are proposed, simply removed from the manuscript.

Previous Reviewer 3 comment #5: The results with N1303K remain the weakest aspect of the work presented here. Although there has been at least one report showing a band of higher molecular mass than band C in cells expressing N1303K-CFTR, there are dozens of papers in which N1303K-CFTR is not processed. At least a comment on this controversial aspect needs to be added. The results are in fact so puzzling that doubt remains if the construct is really bearing N1303K.

We have revised the section concerning N1303K variant data as follows:

'It is interesting that among the 64 variants used in the study, the TrIdx of N1303K is the least similar amongst all cell lines tested to date (0.5 of WT in HEK293 cells compared to 0.1 of WT in HeLa cell or HBE cells (**Fig. S5**)). These results suggest that folding and functional properties of NBD2 encompassing the N1303K SCV cluster could be differentially modulated by the specific biological properties of the tissue in which it is expressed ' (Line 258-264)

In addition, we provided 6 additional references (at the end of this sentence) coming from 6 different laboratories summarizing all the data available on the N1303K variant in different cell lines and contexts which will give a complete overview of the extent variation captured by different laboratories to the reader.

Finally, we have removed some of the highlights on the N1303K variants which were not related to the main take-homes of the paper that might have been confusing.

Previous Reviewer 3 comment #6: I1234V has been shown to affect splicing – thus decreasing considerably the amount of full length CFTR (bearing I1234V) that is synthesized. It doesn't make sense to treat it as a simple missense mutation – and so this mutation should be removed.

We have now removed the I1234V variant from our study. Removal of this variant from our landscape calculations had no impact on any of the conclusions in the paper.